# LARGE-SCALE VIDEO CONTINUAL LEARNING WITH BOOTSTRAPPED COMPRESSION

## ABSTRACT

Continual learning (CL) promises to allow neural networks to learn from continuous streams of inputs, instead of IID (independent and identically distributed) sampling, which requires random access to a full dataset. This would allow for much smaller storage requirements and self-sufficiency of deployed systems that cope with natural distribution shifts, similarly to biological learning. We focus on video CL employing a rehearsal-based approach, which reinforces past samples from a memory buffer. We posit that part of the reason why practical video CL is challenging is the high memory requirements of video, further exacerbated by long-videos and continual streams, which are at odds with the common rehearsal-buffer size constraints. To address this, we propose to use compressed vision, i.e. store video codes (embeddings) instead of raw inputs, and train a video classifier by IID sampling from this rolling buffer. Training a video compressor online (so not depending on any pre-trained networks) means that it is also subject to catastrophic forgetting. We propose a scheme to deal with this forgetting by refreshing video codes, which requires careful decompression with a previous version of the network and recompression with a new one. We expand current video CL benchmarks to large-scale settings, namely EpicKitchens-100 and Kinetics-700, with thousands of relatively long videos, and demonstrate empirically that our video CL method outperforms prior art with a significantly reduced memory footprint.

## 1 INTRODUCTION

Our world evolves endlessly over time. This temporal evolution creates a continuous shift in real-world data distributions. Crucially, resource-constrained autonomous agents must cope with these ongoing changes, akin to humans. Continual learning (CL) offers a practical solution to robustly acquire knowledge in non-stationary environments while amortizing the learning process over the agent's lifespan (Thrun, 1995). In this paper, we focus on CL utilizing long-video understanding to replicate the real-world complexities encountered in actual deployment scenarios. Existing CL research focuses on static images or shorter video clips, thus failing to adequately address the natural shift in data distribution over extended time scales. In this work, we highlight naturally-collected long videos, which we believe is necessary to capture this temporal progression and long-tailedness, properties inherent to online learning. Furthermore, naturally-collected long videos closely align with the principles of human learning scenarios (Damen et al., 2018) that lifelong learning systems aspire to emulate (McCloskey and Cohen, 1989).

The extra temporal axis of video, compared to a static image, can capture rich information such as long-term activities and stories. However, it also brings a few orders of magnitude of more data with the concomitant costs in processing and memory requirements (Han et al., 2022). We highlight that this challenge further compounds in CL systems as they operate over large time scales on a continuous video stream. Additionally, with long videos, CL systems have to mitigate forgetting along a long-range temporal dimension. Consequently, the computational and memory requirements escalate significantly to accommodate these dual constraints, thus necessitating scalable approaches.

In this paper, we propose a memory-based video CL method to learn over naturally-collected long videos. Specifically, our method builds an online video compressor to perform continuous compression and decompression over a neural-code rehearsal buffer, and an online classifier that uses the rehearsal buffer to perform video learning in the compressed space. Different than prior works,

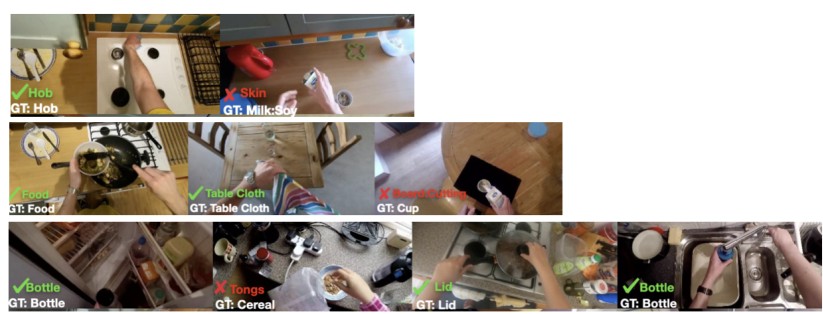

Figure 1: Continual Learning on Epic-Kitchen dataset with noun classification

our rehearsal buffer is neural-code based storing compressed instead of raw RGB-based input. By design, our neural-code rehearsal buffer efficiently handles wide temporal history for rehearsal, necessary to mitigate forgetting in large-scale long video continuous streams.

We draw some inspiration from the internal workings of the mammalian brain and human dreaming, though like most works in CL we cannot claim biological plausibility. Specifically, hippocampal indexing theory states that the hippocampus stores compressed representations of neocortical activity patterns while awake (Teyler and Rudy, 2007; Hayes et al., 2020). Furthermore, the compressed information, also identified as temporal compression of events in episodic memory, enables efficient storage and recall of past experiences (D'Argembeau et al., 2021; Howard, 2018). This phenomenon suggests the significance of temporal compression in efficiently retaining information over long input streams (Jeunehomme et al., 2019), a challenge in video CL. Motivated by this observation, we maintain a compressed temporal buffer. Furthermore, insights from theories in dreaming suggest that human dreams may have evolved to assist generalization and reduce forgetting (Hoel, 2021). The hallucinatory and narrative nature of dreams potentially contribute to refining generative models, enhancing the brain's predictive processing capabilities (where predictions traverse top-down, while sensory input, bottom-up), and improving predictions about future states(Clark, 2013; Hohwy, 2013; Keller and Mrsic-Flogel, 2018; Foulkes and Domhoff, 2014). Inspired by theories about the role of dreaming in learning, we perform continuous compression and decompression, emulating a bottom-up and top-down approach that reinforces the stability of representations. We note that this inspiration does not make current neural networks biologically plausible, as they rely on back-propagation for learning, which is not supported by biological evidence (Crick, 1989; Lillicrap et al., 2016; Whittington JC, 2019).

In this work, we focus on two broad settings in CL, and evaluate our method under both. The first is incremental learning – training a network from scratch by presenting it with a sequence of disjoint data distributions. This models a shifting data distribution as a sequence of distributions (Chaudhry et al., 2019; Rebuffi et al., 2017; Lopez-Paz and Ranzato, 2017). This closely mimics biological learning, i.e. an agent learning solely from sequential experience. A variation of incremental learning is to allow an initial pre-training phase (Douillard et al., 2020), where the network is trained on a large subset of the classes (e.g. half of them) non-sequentially (independently and identically distributed, IID), and then it is incrementally adapted. This setting more closely follows the common usage of ML models, where usually there is at least some relevant dataset for pre-training before deploying a system, and can circumvent many challenges posed by the incremental learning setting, such as computational cost and representational drift.

Our key contributions are as follows:

1. A neural-code memory-based video continual learning framework that operates on large-scale long videos.

2. A code refreshing scheme that minimizes representation drift in a buffer of codes that were initially created with different versions of the same compressor.

3. An evaluation of video CL in large-scale video datasets, namely Epic-Kitchens-100 and Kinetics-700.

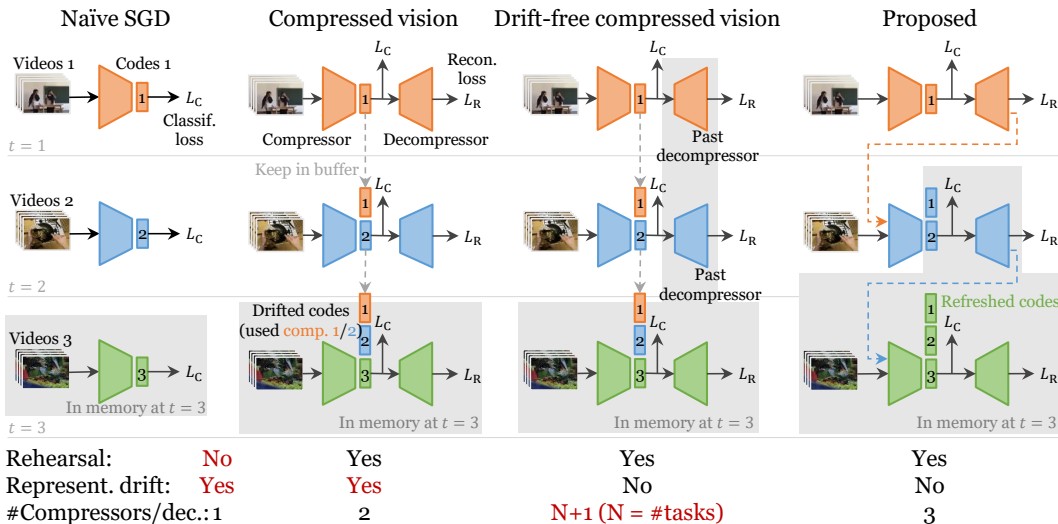

Figure 2: Overview of the differences between our proposed scheme and alternative compressed buffer strategies. Using a compressed buffer for rehearsal (column 2) risks representation drift, since codes were created with a different version of the trained encoder (represented as 3 different colors). Decoding without drift requires snapshots of the decoder over time (column 3), but the memory growth is unbounded. Our proposed scheme (column 4) refreshes codes to keep them from drifting, while only requiring a single snapshot of the last decoder.

4. Empirical evaluations of our method in both datasets, in 2 popular CL settings: with pre-training, and incrementally from scratch.

We evaluate our framework for noun and action classification task on Epic-Kitchens-100 and Kinetics-700 datasets respectively. Our method significantly outperforms state-of-the-art performance under both the settings. We believe that this is the first work to extend continual learning to large-scale naturally-collected long videos.

## 2 RELATED WORK

**Continual Learning with Images and Videos.** Most current CL systems show promising results in the image domain, which primarily involves artificially-constructed sequences of images and transfer of declarative knowledge of entities and concepts (Buzzega et al., 2020; Qu et al., 2021; Lopez-Paz and Ranzato, 2017). Different than these, our focus on naturally-collected long videos creates a continuous data distribution shift and serves as a robust test-bed for evaluating CL systems under real-world task settings that require the transfer of procedural knowledge over extensive time spans. Natural videos simulate real-world conditions, such as the nuanced understanding of actions or behaviors in long video sequences (Damen et al., 2018). Furthermore, the deployment of CL systems in real-world settings, like surveillance cameras or autonomous vehicles, necessitates their ability to effectively learn from continuous long video streams over significant time scales(Doshi and Yilmaz, 2022; 2020). There have been some works in CL that operate on videos, however, are limited to processing only few-seconds to minutes long videos or do not propose scalable approaches to tackle the high memory and computational requirements. OAK (Wanderlust) (Wang et al., 2021) released a benchmark with long ego-centric videos but was limited to testing current CL algorithms with a narrow task domain focused on coarse-grained object detection with sparse annotations. This benchmark was also used in Efficient-CLS (Wu et al., 2023) which proposed a slow-fast CL method with an episodic memory similar to (Rebuffi et al., 2017; Lopez-Paz and Ranzato, 2017; Chaudhry et al., 2019). With its focus on Complementary Learning Systems (Kumaran et al., 2016), Efficient-CLS (Wu et al., 2023) is complementary to other CL methods, augmenting them with a pair of slow and fast learners, and using the former to generate pseudo-

labels for the later. (Wu et al., 2023) also shows performance on EgoObjects (Zhu et al., 2023), a fine-grained ego-centric dataset with seconds-long clips. This contrasts with our experiments using Epic-Kitchens-100 (Damen et al., 2018), with minutes to hours-long videos. CLAD (Verwimp et al., 2022), a CL benchmark for autonomous driving, repurposed an image dataset to form a temporal stream. It proposed a single (days-long) "video" (time lapse sequence of images), introducing domain shifts at different frequencies (*e.g.* time, location, different objects, viewpoint). While having a single very long video is a reasonable axis to expand video CL evaluation, we extend it to (Damen et al., 2018) with thousands of videos, each minutes to hours-long. In addition to location, time, objects, viewpoints, (Damen et al., 2018) also poses domain shifts resulting from fine-grained human-object interactions and cinematography changes, thus distinguishing it from (Verwimp et al., 2022). To the best of our knowledge, we are the first to build a practical CL algorithm in a large-scale long video setting, and thoroughly evaluate it in a realistic setting.

**Memory-Based Continual Learning.** Memory-based algorithms have demonstrated strong performance in CL (Saha and Roy, 2021; Prabhu et al., 2020; Chaudhry et al., 2019). During training, a memory buffer stores data instances from the past and rehearses them while training new tasks in order to consolidate previously learned knowledge to mitigate catastrophic forgetting. (Hayes et al., 2020) proposed a compression-based CL method over static images and natural language, however, did not address challenges arising from CL over long videos. Furthermore, most current research primarily shows the relevance of different memory budgets, balancing or rehearsal techniques (Prabhu et al., 2020). While we don't argue whether an unbounded or bounded memory budget is beneficial, we show that under any budget, compression leads to significant gains.

**Video Compression.** Training robust video representations has proven to be more challenging than learning deep image representations, due to the enormous size of raw video streams and the high temporal redundancy. Superfluous information can be reduced by up to two orders of magnitude by video compression (Wu et al., 2018; Wiles et al., 2022). Importantly, compressed video representation has a higher information density, and additionally the training is made easier, as generic features are already extracted. The signals in a compressed video provide free, albeit noisy, motion information (Li et al., 2023; Wu et al., 2018). In video learning, it remains a challenge how to accurately capture key information, and several works have tried techniques such as token dropout, frame sampling and key information detection (Yan et al., 2020; Han et al., 2022; Zhi et al., 2021). Compression on the other hand presents an elegant solution for these challenges (Wu et al., 2018).

**Robot Lifelong Learning.** A strand of robotics delves into continual learning methodologies utilizing videos and feedback mechanisms. In this realm, robots are tasked with acquiring and refining their skills and knowledge over time (Thrun, 1995; Liu et al., 2021; 2023). Robot lifelong learning typically focuses on active learning and the effect of an agent's actions in the environment.

## 3 BACKGROUND

### 3.1 COMPRESSED VISION

Our method builds on compressed vision, proposed by Wiles *et al.* (Wiles et al., 2022). The main concept is to train any classifier on small codes (embeddings) obtained from video frames, instead of the frames directly. By using a frozen compressor network to obtain the codes, and performing data augmentation (to avoid overfitting) directly in the code latent space instead of the input space, they can store extremely long videos in memory compared to traditional approaches. Their pipeline consists of three training phases. 1) They train a *neural compressor* $c = (\phi, \psi)$, where $\phi$ and $\psi$ denotes the encoder and decoder respectively, using a VQ-VAE (Van Den Oord and Vinyals, 2017). $c$ takes videos $X$ as input and produces neural codes $x \in \mathbb{R}^{s \times h \times w}$. 2) They train an augmenter network $a$, that takes as input $x$ and predicts codes $\hat{x}_i$ that correspond to randomly-transformed video frames. 3) Lastly, they train a video task classifier that takes as input $\hat{x}$ to solve a given downstream task, and prevent over-fitting by using $a$ to perform data augmentation directly in the space of the codes. Note that in the first phase, once $c$ is trained, $x \in X$ are stored in a buffer, $c$ is frozen and the original videos are no longer needed. Wiles *et al.* (Wiles et al., 2022) show strong performance results (under 5% drop) at high compression rates ($256\times$ and $475\times$).

## 3.2 Incremental Learning

A common scenario in CL (Chaudhry et al., 2019; Rebuffi et al., 2017; Lopez-Paz and Ranzato, 2017) is incremental learning – training a network by presenting it with a sequence of $n$ tasks consisting of disjoint data distributions, sequentially, as $T = \{t_i\}_i^n$. This models a shifting data distribution as a sequence of distributions. Concretely, a learning model observes a *continuum of data*, which is a concatenation of $m$ samples from each of the tasks, for a total of $nm$ samples, as follows:

$$D = \{x_{j,i}, y_{j,i}\}_{j,i}^{m,n} \tag{1}$$

$$x_{j,i} \overset{iid}{\sim} X_{t_i}, \quad y_{j,i} \overset{iid}{\sim} Y_{t_i} \tag{2}$$

$X_{t_i}$ is a distribution over images for task $t_i$, and $Y_{t_i}$ is a distribution over its target vectors (for example, action classes). For simplicity, we assume that the continuum samples are IID within a task.

The main advantage of this setting is that it represents the most stringent test of continual learning, by training from scratch. It also more closely mimics biological learning, i.e. an agent learning solely from sequential experience.

## 3.3 Pre-training and Incremental Learning

A variation of incremental learning is to allow an initial pre-training phase (Douillard et al., 2020), where the network is trained on a large subset of the classes (e.g. half of them) IID, and then is incrementally adapted as before. This more closely follows the common usage of ML models, where usually there is at least some relevant dataset for pre-training before deploying a system.

# 4 Method

Similarly to Sec. 3, we aim to train a deep neural network by presenting it with a sequence of $n$ tasks of disjoint data distributions, i.e. eq. 1. The main difference is that each $x_{j,i}$ is a video clip, and each $y_{j,i}$ is now a video class (e.g. a human action label).

## 4.1 The Ideal Case: IID Sampling

We will first present the ideal case, where a learner has access to all available samples, sampled IID. This avoids catastrophic forgetting and allows us to introduce the concepts in a simplified form. We aim to train a feature extractor or compressor $c = (\phi, \psi)$, composed of an encoder $\phi$ and decoder $\psi$, as well as a classifier $q$ which takes the features from the encoder. The objective of the compressor, trained on the full dataset from eq. 1, is defined as:

$$\psi^*, \phi^* = \arg\min_{\psi,\phi} \left( \mathbb{E}_{t_i \sim T} \mathbb{E}_{x_j \sim X_{t_i}} \left( ||\psi(\phi(x_j)) - x_j||^2 \right) \right). \tag{3}$$

The classifier is simply trained with a cross-entropy loss $L$ for classification (or another loss for a different downstream task):

$$q^* = \arg\min_{q} \mathbb{E}_{t_i \sim T} \left( \mathbb{E}_{(x_j,y_j) \sim (X_{t_i}, Y_{t_i})} L(q(\phi(x_j)), y_j) \right) \tag{4}$$

This is, of course, an idealized situation where it is possible to have random access to any sample. Next we'll turn to the CL scenario where we are given only a single task (time) $t_i$ at a time, and cannot directly access past samples.

## 4.2 Incremental Learning

In this setting, we train the compressor continually with new classes. It suffers from forgetting if the old classes are not represented, so we employ a rehearsal strategy while training the compressor. Unlike (Wiles et al., 2022), in Setting 2 as time progresses, the compressor observes new data samples unseen during past tasks. Additionally, during any task $t_k$ described in equation 1, the

Figure 3: Overview of the proposed compressed continual learning pipeline. Our method trains a video compressor as an autoencoder, together with a classifier, while storing short compressed codes describing the videos in a buffer for rehearsal of past samples. Our method continually refreshes codes from past tasks $t-1$ so that they work with the compressor for the current task $t$, ensuring the stability of the representations over time.

learner receives video clip frames that are never revisited, creating a challenge for gradient-descent-based learning. As the compressor $c$ is also learning (and changing) as time progresses, how do we adapt it to the shifting video distribution?

### 4.2.1 REHEARSAL BUFFER AND TEMPORAL EVOLUTION OF MODELS.

Because we will train a model sequentially over the tasks, and it will be different for each task, we need to consider a *sequence of models* $(c_1, q_1), \ldots, (c_n, q_n)$, one per task $t_i$.

In order to allow training on past samples, so that the loss value on them is maintained, some form of memory (explicit or implicit) is also required. In this work we maintain a buffer denoted as $B_{i-1}$. At time $t_i$ it is defined as

$$B_{i-1} = \{e_{j,k}\}_{j,k}^{m,i-1}, \quad e_{j,k} = \phi_{t_{i-1}}(x_{j,k}) \tag{5}$$

where $k$ iterates over previous tasks (1 to $i-1$), $j$ iterates over samples per task (1 to $m$), and $e_{j,i}$ denotes the compressed video clip. The *neural-codes* based buffer $B_{i-1}$ contains previously observed video codes necessary to maintain old concepts from prior tasks. During the task $t_i$, when training $c_i$ and $q_i$, we only have access to the last state of the buffer $B_{i-1}$ and video examples from the current task, $X_{t_i}$.

### 4.2.2 INCREMENTAL LEARNING FORMULATION.

Let us consider the first task. Adapting eq. 3 to focus on the first task, we have:

$$\psi_1^*, \phi_1^* = \arg\min_{\psi_1, \phi_1} \left( \mathop{\mathbb{E}}_{x_j \sim X_{t_1}} \left( ||\psi_1(\phi_1(x_j)) - x_j||^2 \right) \right), \tag{6}$$

and an identical adaptation for the classifier from eq. 4. Similarly, for the second task, we have the loss equation:

$$\psi_2^*, \phi_2^* = \arg\min_{\psi_2, \phi_2} \left( \mathop{\mathbb{E}}_{x_j \sim X_{t_2}} \left( ||\psi_2(\phi_2(x_j)) - x_j||^2 \right) + \tag{7}$$

$$\mathop{\mathbb{E}}_{e_j \sim B_1} \left( ||\psi_2(\phi_2(s_j)) - s_j||^2 \right) \right) \tag{8}$$

$$\text{where } s_j = \psi_1(e_j) \tag{9}$$

| Setting | Method | Kinetics-700 (K-700) | | | EpicKitchens-100 (EK-100) | | |
|---|---|---|---|---|---|---|---|
| | | Train. ↑ | Eval. ↑ | AvgF ↓ | Train. ↑ | Eval. ↑ | AvgF ↓ |
| Pretraining | Upper Bound | 57.10 | 48.20 | – | 42.10 | 35.90 | – |
| | BootstrapCL (Ours) | **56.25** | **46.50** | **5.50** | **40.10** | **33.20** | **9.70** |
| | REMIND Hayes et al. (2020) | 43.51 | 35.90 | 49.20 | 30.89 | 24.60 | 56.3 |
| Incremental | Upper Bound | 48.20 | 44.10 | – | 36.20 | 32.0 | – |
| | BootstrapCL (Ours) | **44.60** | **38.80** | **15.20** | **32.60** | **28.10** | **21.60** |
| | SMILE Alssum et al. (2023) | 40.56 | 29.20 | 62.50 | 28.71 | 19.20 | 67.8 |
| | vCLIMB Villa et al. (2022) | 39.12 | 28.65 | 65.10 | 27.11 | 18.5 | 66.5 |
| | GDumb Prabhu et al. (2020) | 37.61 | 18.70 | 52.40 | 25.30 | 15.60 | 60.10 |

Table 1: Comparison of our method and baselines (average training (Train) and evaluation accuracy (Eval), and average forgetting (AvgF)), on K-700 and EK-100, with pre-training and incremental settings (as described in Sec 5.2 and 5.3). We set 654 Mb (in K-700) and 714 Mb (in EK-100) as the maximum memory budget for our method and baseline experiments above (as described in Sec 5.4). Upper Bound refers to the upper bound baseline which has unbounded memory budget (described in Sec 5.4).

where the first expectation is over the current batch, and the second expectation is over codes stored in the buffer, which are decoded by $\phi_1$. Note that it is important to decompress the buffer using the decoder parameters from the previous task $\psi_1$, not the one currently being trained $\psi_2$, in order to be consistent with the encoder they were compressed with, $\phi_1$.

As for the classification objective (eq. 4), it is also adapted using a mix of codes from the buffer and from the batch of samples in the current task:

$$q_2^* = \arg\min_q \left( \mathbb{E}_{(x_j,y_j) \sim (X_{t_2}, Y_{t_2})} L(q(\phi_2(x_j)), y_j) + \mathbb{E}_{(e_j,y_j) \sim B_1} L(q(\phi_2(s_j)), y_j) \right), \quad (10)$$

where we reuse eq. 9, and slightly abuse notation to retrieve the classification label $y_j$ associated with the buffer's code $e_j$.

We can apply equations 6 and 7 recursively to any task $t_k$ by using the buffer and compressors from the respective tasks, and thus extend it by induction. Fig. 3 gives an overview of this process.

### 4.3 CONTINUAL LEARNING WITH PRE-TRAINING

Another natural setting as illustrated in (Douillard et al., 2020) is to consider networks that undergo pre-training with IID samples prior to incremental learning. In this setting, ~~we have two phases.~~ there are two phases. In the first phase, the model is pre-trained with half of the dataset's classes and in the second phase, the model is incrementally trained with rest of the classes.
Following (Douillard et al., 2020)'s protocol, in the first phase we ~~first~~ pre-train the compressor and classifier with half of the dataset's classes, and in the second phase, incrementally train the classifier as in sec. 4.2.2.
Note that an important distinction from the previous setting described in 4.2.2 is that after phase 1 finishes, we can freeze the compressor – assuming that the pre-training is sufficient to learn relevant features – and as a result, during phase 2, we do not decompress our buffered codes. This avoids representation drift of the codes and simplifies the method, which does not need to back-propagate through the codes.

It is interesting to contrast this pre-training setting to the incremental learning only setting (sec. 4.2.2). Continuously decompressing and adapting the codes incurs a computational cost and risks representational drift. Under a bounded memory budget, the compressor may be under-trained and fail to produce robust codes. The pre-training setting circumvents these issues, while still enjoying the benefits of incremental learning of downstream tasks.

# 5 EXPERIMENTS

To demonstrate our method empirically, we evaluate on video-based CL baseline and propose an extension of image-based CL evaluations to large-scale video datasets. We use Kinetics-700 (K-700) (Kay et al., 2017) and Epic-Kitchens-100 (EK-100) (Damen et al., 2018), where we perform action and noun classification tasks respectively.

## 5.1 IMPLEMENTATION DETAILS

We use the same compressor architecture as Wiles *et al.* (Wiles et al., 2022), which is based on a ResNet, and refer the reader to their work for a complete review. Compressor training differs in the two settings as described below. In both the settings, we maintain a queue for the rehearsal buffer to store the video codes. For the downstream video task classifier, we use S3D (Xie et al., 2018) for K-700, and short-term S3D for EK-100, which takes the compressed codes as inputs. We follow the specifications from Wiles *et al.* (Wiles et al., 2022) to adapt the network's kernel size and stride at every layer. We experimented with different architectures for the classification task, in order to find the optimal settings (further results in appendix A). We use compression rate $256\times$ unless stated otherwise. We apply random horizontal flipping and random cropping of size $224 \times 224$ from frames resized such that the short side $\in [256, 340]$ as data augmentation. Each video clip of dimensions $\sim 224 \times 224 \times 14$ $224 \times 224 \times 3 \times 32$ (32 RGB frames) corresponds roughly to a compressed code of size 0.0013 Mb.

## 5.2 SETTING: CONTINUAL LEARNING WITH PRE-TRAINING

**Dataset.** Following the experimental protocol in Douillard et al. (2020), we split K-700 into 2 parts. The first split consists of Kinetics-400 (K-400), and the second split contains the remaining 300 classes of K-700. Similarly, we split EK-100 into 2 parts, the first with 17 participants and the second with 16 participants. Classes are sampled IID in the first dataset split respectively. For K-700, the second split has 10 tasks with 30 non-overlapping classes per task. For EK-100, the second split has 17 tasks with 1 participant per task. Videos are sampled IID within every task.

**Training.** As described in Section 3, this setting has two phases. In the first phase, we follow IID training. We train the compressor $c$ and classifier for 300 epochs with a batch size of 32, and use the Adam optimizer with learning rate of $0.01$ and weight decay of $10^{-5}$. $c$ is frozen at the end of pre-training. We store the compressed codes into the queue for all the classes in this phase, and then train the classifier with these stored codes. We start the second phase with the pre-trained classifier from the first phase and train it incrementally over 10 tasks for K-700 and 17 tasks for EK-100. We pass the transformed video inputs through the frozen compressor, store the resulting codes into the queue and use them as inputs for the classifier. We receive new class samples at every task, and assume IID sampling over those. We train the classifier for 2 epochs with the compressed codes corresponding to new samples and those stored in the rehearsal buffer. Note that the buffer also includes the codes from pre-training classes, plus from all tasks seen so far. We also perform ablations varying the number of epochs per task and class splits. The incremental training over the classifier completes once all the tasks are processed.

## 5.3 SETTING: INCREMENTAL LEARNING FROM SCRATCH

**Dataset.** For K-700, we have 35 tasks with 20 non-overlapping classes per task. For EK-100, we have 33 tasks with video samples from 1 participant per task. Videos are sampled IID within every task.

**Training.** We train the compressor and classifier incrementally, and within each task, we follow IID training. So, we first train the compressor for 1 epoch and then the classifier for 30 epochs unless stated otherwise. To train the compressor, we use a batch size of 16, and Adam optimizer with learning rate of $0.01$ and weight decay of $10^{-5}$. At every task, during compressor training, we decompress compressed codes from the buffer (unless empty) using the latest compressor, and obtain the corresponding RGB values. We then re-train the compressor jointly with the decompressed codes and new samples. Lastly, we store the freshly compressed codes into the buffer, and freeze the compressor for that task. We use the stored codes from the current and past tasks as inputs to the

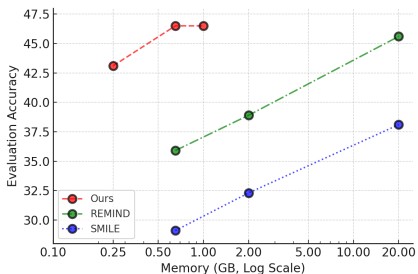

Figure 4: Average evaluation accuracy for different methods, with varying memory budgets on Kinetics-700.

| Dataset | Memory (MB) | BootstrapCL (Ours) | RGB buffer |
|---|---|---|---|
| | Buffer | 654 | $1503 \times 10^3$ |
| Kinetics-700 | Models | 750 | 250 |
| | Total | 1404 | $1503.2 \times 10^3$ |
| | Buffer | 714 | $1640 \times 10^3$ |
| EpicKitchens-100 | Models | 750 | 250 |
| | Total | 1464 | $1640.2 \times 10^3$ |

Table 2: Memory footprint of our method with a compression ratio of $256\times$ versus a traditional buffer of RGB images.

task classifier. We store the resulting ~~the~~ codes for every video clip into the queue, and freeze the compressor for that task. So, at every task, we interleave between compressor and classifier training. This training process is repeated for the total number of tasks.

## 5.4 BASELINES

Our method lies at the intersection of memory-based and video CL. For memory-based CL, we compare with GDumb (Prabhu et al., 2020) and REMIND (Hayes et al., 2020) which focused on image-based analysis. For video CL, we compare with SMILE (Alssum et al., 2023), which is also a memory-based CL method. We also design an upper bound baseline using an unbounded RGB memory budget. To compare with REMIND (Hayes et al., 2020), we use our pre-training set-up, as these baselines rely on a pre-trained architecture. For the rest of the baselines, we use our incremental learning set-up. For further details on video samples storage, please refer to Appendix A.

In our baseline comparisons for K-700 and EK-100, we set 654 Mb and 714 Mb respectively for the maximum memory budget of all methods, in order to ensure a fair comparison. These values were chosen as the maximum memory that our method requires, and they are well within the capacity of modern hardware. For some baselines, we also show comparisons with different memory budgets in the ablations section. During the incremental learning phase, at every task, we split the storage space equally for each past task up to the buffer limit. Denote $K$ as the total number of video samples that can be stored under the assigned memory budget. Then the total number of samples from each past task at the $n^{th}$ task in the incremental learning setting is given by $K\frac{1}{n-1}$. The total number of samples from each past task at the $n^{th}$ task in the pre-training setting, where we add one task for the pre-training phase, is $\frac{K}{n}$.

## 5.5 EVALUATION AND METRICS

We report the average accuracy (Lomonaco et al., 2021) after training and evaluation, and average forgetting (Lomonaco et al., 2021) after evaluation for our method and baselines in Table 1. The average accuracy is the average on all the tasks measured at the conclusion of the task sequence. We show some examples of our method's predictions, learned over time, in Fig. 1. We report the total memory buffer size and its equivalent size when storing raw pixel frames in Table 2, and show ablations with a different compression rate in Appendix A.

## 6 RESULTS AND ANALYSIS

### 6.1 PROPOSED METHOD

We find that our method outperforms the baselines and achieves average accuracy comparable to the upper bound baseline in both our proposed settings, as seen in Table 1. We observe that our pre-trained compressor captures class-agnostic semantics effectively. For samples unseen during the pre-training phase, it outputs robust compressed codes without further training, thus enabling the online

classifier to achieve strong performance. In incremental learning only setting, at every successive task, since our method decompresses and rehearses past codes, it learns to jointly represent the features for both old and new tasks. This allows it to output robust codes for downstream video application. Due to the highly efficient memory, it enjoys full rehearsal of samples from all past tasks, thus our classifier can efficiently represent all classes, and achieve strong performance. Our compression strategy is well-optimized such that, even for very large number of samples ($> 500K$ video samples) with high memory footprint, we only need a small amount of memory ($< 2\,\text{GB}$). One interesting finding from our work is that we do not need to apply any frame selection or sampling strategy, even for very large videos.

### 6.2 State-of-the-Art Methods

**Memory-Based CL Baselines.** We see that GDumb Prabhu et al. (2020) suffers from catastrophic forgetting, as the evaluation accuracy is significantly lower than our method. This is due to lack of sufficient samples for rehearsal. This also shows that the strongest rehearsal-based technique is unable to cope with the high memory requirements for videos. Similar to GDumb, which also compares with other rehearsal-based works such as Saha and Roy (2021); Prabhu et al. (2020); Chaudhry et al. (2019); Alssum et al. (2023), these assume RGB values stored in the buffer, however, an unbounded budget is unfeasible in practise. Therefore they further limit the budget by employing different sampling strategies, resulting in performance degradation.

Our method with a 20x higher compression rate outperforms REMIND Hayes et al. (2020), a compression-based CL technique. As a result, our memory buffer maintains a wider temporal history compared to theirs and delivers a greater performance accuracy on both the datasets. Furthermore, they do not refresh representations instead only the final layer features, which may explain the lower performance on downstream applications.

**Video CL Baselines.** We observe that SMILE Alssum et al. (2023) requires a large memory budget to meet the state-of-the-art performance as seen in their work. From Table 1, we see that their performance degrades significantly under both datasets under bounded the memory budget. Furthermore, in the case of long videos, as dense temporal sampling is necessary for maintaining temporal association and long-term context to benefit inference Han et al. (2020), their performance further degrades as they perform significant temporal down-sampling.

### 6.3 Ablation Experiments

We also describe and report the average evaluation accuracy under various memory budgets for our method and baselines in Fig 3. We report results for both higher and lower memory budgets. We can see from this performance memory plot that our method requires significantly less memory to achieve strong performance compared to prior art. Our method's memory budget is well within the capacity of modern hardware. We also describe and report results for further ablations in Appendix A.

## 7 Conclusion

In this work we presented a method to perform continual learning over long-videos, mitigating catastrophic forgetting. Video CL poses considerable challenges, one of them being the high memory requirements. We propose to use compressed vision as a way to increase substantially the buffer size used for rehearsal in CL, and highlight the need to devise an appropriate strategy to deal with the representation drift of the compressor (i.e. codes become stale compared to the most recent compressor state). We demonstrate encouraging results in 2 large-scale video datasets, Epic-Kitchens-100 and Kinetics-700. We also study 2 different settings of CL, with pre-training and from scratch. We believe that compressed vision can play an important role in scaling up methodologies developed for images and adapt them to videos. In future work we would like to explore even more long-duration videos, and other tasks that go beyond action classification.

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

## A APPENDIX

### A.1 VIDEO DATASET COMPARISONS

### A.2 DATASET DETAILS

**Epic-Kitchens-100** The average video length is 20 minutes, longest video length is 1.5 hours and shortest video length is 5 minutes. Total video footage length is 100 hours. Each video is at 25 frames per second. We further describe the dataset annotations. Each video is associated with a participant and video identifier. Each video is split into a block of frames (segment) with a start and a stop timestamp, and indicated with the start and stop frame. A video segment is labeled with all the noun categories present in it (so multiple labels per clip). The labeling is at the video segment level. There are a total of 331 noun classes covering various nouns involved in kitchen actions (including everyday equipment). Smooth transitions between classes are ensured by presenting the segments to the models chronologically.

**Kinetics-700** The average video length is 10 seconds, longest video length is 15 seconds and shortest video length is 7 seconds. Each video is at 25 frames per second. There are 700 classes in total, and each class is also associated with an integer label (which is an integer value from 0 to 699). Each video is associated with a class label.

| Dataset | Longest Video Length (secs) | Average Video Length (secs) | # of Object / Action Categories | Video understanding Setting | Used In |
|---|---|---|---|---|---|
| ActivityNet | 600 (10 mins) | 120 | 203 | short | SMILE, vCLIMB, DPAT |
| Kinetics (400/600/700) | 20 | 10 | 400 / 600 / 700 | short | SMILE, vCLIMB, Ours |
| UCF101 | 8 | 5-7 | 101 | short | ST-Prompt, FrameMaker, SMILE |
| HMDB51 | 6 | 6 | 51 | short | ST-Prompt, FrameMaker |
| Something-Something V2 | 6 | 4-6 | 174 | short, fine-grained | FrameMaker, ST-Prompt |
| Epic-Kitchens-100 | 5400 (1.5 hrs) | 900-1200 (15-20 mins) | 331 | long, fine-grained | DPAT (concurrent work), Ours |

Table 3: Summary of video datasets: The following table describes each video dataset with the length of its longest video (column 2), average length (column 3), classification and temporal complexity in its video understanding setting (column 4, 5), and the respective CL works these datasets are used in (column 6).

| participant id | video id | start time | stop time | nouns | noun_classes |
|---|---|---|---|---|---|
| P01 | P01_01 | 00:29.22 | 00:31.32 | ['fridge'] | [12] |
| P01 | P01_01 | 09:07.40 | 09:09.01 | ['container', 'fridge'] | [21, 12] |
| P01 | P01_105 | 00:27.01 | 00:27.83 | ['container', 'cupboard'] | [21, 3] |
| P02 | P02_108 | 00:43.83 | 00:45.92 | ['biscuit', 'cupboard'] | [104, 3] |

Table 4: Example annotations from EK-100 dataset

| label | youtube_id | start_time | stop_time |
|---|---|---|---|
| 'baking cookies' | JJWwLganiil | 31 | 41 |
| 'gymnastics tumbling' | 5KbfOS44-gM | 49 | 59 |
| 'writing' | iYcARQA6VIU | 0 | 10 |
| 'wrapping present' | Qo5lspgmqPU | 167 | 177 |

Table 5: Example annotations from K-700 dataset

## A.3 AVERAGE FORGETTING METRIC (AVGF)

Let $a_{i,t}$ be accuracy on task i of the model that was trained on $t$ tasks, where $i < t$. Average forgetting measures how much performance has degraded across the first $t-1$ tasks. To do so, this metric uses the difference between best-obtained performance of the desired task and the performance obtained from the current incremental learner.

$$F_t = \frac{1}{t-1} \sum_{1}^{t-1} f_{i,t} \quad \text{where} \quad f_{i,t} = \max_{q<t}(a_{i,q} - a_{i,t}) \quad \text{or} \quad f_{i,t} = a_{i,i} - a_{i,t} \tag{11}$$

## A.4 BASELINE DETAILS

GDumb (Prabhu et al., 2020) maintains a randomly-sampled RGB memory buffer. It stores all samples until the buffer is full and then stops storing. We store approximately 226 and 490 video samples respectively for K-700 and EK-100 in the buffer. So, for K-700, for incremental setting, if $n = 35$, we have 6 samples from each past task rounding down. In the pre-training setting, if

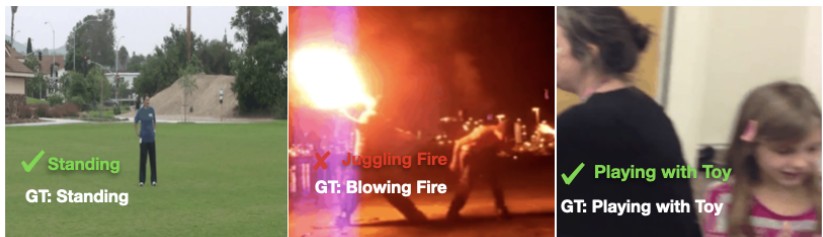

Figure 5: Continual Learning on Kinetics-700 dataset with action classification.

$n = 10$, 22 samples respectively. And, for EK-100, if $n = 33$, we roughly have 15 samples from each past task, and if $n = 16$, 30 samples respectively.

REMIND (Hayes et al., 2020) proposes a compression technique using a two-stage process. In the first stage, it compresses the current input. This stage is analogous to the compression phase in our method. In the second-stage, it reconstructs a subset of previously compressed representations, and mixes them with the current input. It then updates the plastic weights of the network with this mixture. The second stage is analogous to decompression phase and rehearsal in our method to maintain stability of learned and new input representation.

For REMIND (Hayes et al., 2020), we can store approximately 29K and 77K video samples respectively for K-700 and EK-100 in the buffer. For K-700, as $n = 35$, we have 830 samples from each task rounding down. And, for EK-100, $n = 33$, we roughly have 2.3K samples from each task. We directly apply their method by operating on RGB frames from videos instead of RGB samples from images. For base initialization phase, we use 20 classes for K-700 and 1 participant for EK-100 adapting their protocol as on ImageNet

SMILE (Alssum et al., 2023) introduces a memory-based video CL baseline that maximizes the memory buffer usage by storing a single RGB frame per video. To combat the distribution shift between real video clips per CL task and in-memory images (represented as boring videos(Carreira and Zisserman., 2018)), SMILE introduces a secondary loss. The method favors diversity of videos over temporal data per video. Their single-frame memory allows to directly apply image-based CL methods to the video domain. Similar to observations in GDumb (Prabhu et al., 2020), SMILE (Alssum et al., 2023) also reports strong performance with a random sampling technique.

For (Alssum et al., 2023), We store approximately 3164 and 6860 unique video samples respectively for K-700 and EK-100 in the memory buffer. We use the SMILE+BiC baseline (Alssum et al., 2023) (as it gives their stronger performance on Kinetics). We use our incremental setting for comparison as it is similar to their proposed set-up. For K-700, if $n = 35$, we roughly have 24 samples from each past task. And, for EK-100, if $n = 33$, we roughly have 210 samples from each past task.

### A.5    ABLATION EXPERIMENTS

We report ablations with a different compression rate in Table 3. We report ablations with 40 epochs per task in Table 4, (different than 30 epochs used in our main experiments) which shows a slight performance increase. This can be attributed to longer network training in the IID phase per task which allows for further loss reduction. We also show ablation with a new split for classes per task in Table 5. For K-700, we try with 15 classes per task for 45 tasks in incremental setting, and 20 classes per task for 15 tasks in pre-training setting. For incremental setting, the training accuracy slightly increases due to fewer classes per task, however, the evaluation accuracy also reduces, indicating possible over-fitting. We see minimal effect in the pre-training setting, possibly due to stable class-agnostic representations learned during pre-training phase.

| Compression | Kinetics-700 | | EpicKitchens-100 | |
| Setting | Train. | Eval. | Train. | Eval. |
|---|---|---|---|---|
| Pretraining | 56.9 | 47.4 | 41.0 | 34.5 |
| Incremental | 46.0 | 40.1 | 33.6 | 29.0 |

Table 6: Our method with a different compression rate ($50\times$). We report training (Train) and evaluation (Eval) performance.

| | Kinetics-700 | | EpicKitchens-100 | |
| | Train. | Eval. | Train. | Eval. |
|---|---|---|---|---|
| Pretraining | 56.8 | 47.9 | 41.4 | 35.5 |
| Incremental | 47.2 | 41.9 | 34.7 | 31.1 |

Table 7: Our method's ablation with a different number of trainin epochs (40). Training (Train) and evaluation (Eval) performance reported above

| Setting | Method | Kinetics-700 | |
| | | Train. | Eval. |
|---|---|---|---|
| Pre-training | BootstrapCL (Ours) | 56.8 | 47.0 |
| Incremental | BootstrapCL (Ours) | 46.6 | 36.1 |

Table 8: Our method's ablation with a different split as explained in equation A.5. Training (Train) and evaluation (Eval) performance are reported above.