# OpenReview forum: "Large Scale Video Continual Learning with Bootstrapped Compression"
_ICLR.cc/2025/Conference — Submitted to ICLR 2025_

### Official Review · Reviewer_ngdC · 2024-11-03

**Soundness:** 2
**Presentation:** 2
**Contribution:** 2
**Rating:** 3
**Confidence:** 3

**Summary:**

This paper presents a continual learning (CL) framework for video. The proposed method (pre)trains a compressor for video frames with an encoder and decoder. Additionally it maintains a buffer of past codes which are used when changing task. The system uses these buffers to do, in the case of experiments in the paper, noun and action classification. Catastrophic forgetting is minimized by maintaining the previous task buffer and making sure the compressor doesn't drift too much when changing tasks.

**Strengths:**

Originality:
While most of the work is based on existing literature, the use of compressed representations in this context is novel.

Quality:
It's nice to see some "real world" datasets being used in this context so there is a beginning of good experimental validation here (but see below). The ablations in the appendix should have been in the main paper, but are nice.

Clarity:
The paper is nicely structure but see below.

**Weaknesses:**

Unfortunately the paper suffers from several weaknesses;

Experimental validation - while I appreciate the use of real world video the experimental validation is lacking. There are only two tasks used and if a method is aiming to show improvement in continual learning then I would really expect more. For example including more datasets (Ego4D, SSv2 for example) and more tasks (dense tasks, pixel prediction) would have made the case of the paper stronger.

Analysis - there is very little analysis as to what the model learns and how - the main ablation is the previous task buffer size, the rest is in the appendix but not a lot of analysis of the significance of the results is given. I would have loved to see how the compressed representation evolve as more tasks are introduced - do they stay the same? do they change abruptly to fit the new task (while still being meaningful for the old ones)? some visualization of the learned representation would be nice as well.

Clarity - I found the paper hard to follow. The model and problem set up are not well explained and the figure captions do little to help. Specifically, the method section (4) needs more context with clear definition of what tasks are and how they evolve over time. Figure 2 caption should be extended - the model is quite simple (I think) and should be completely understandable from that figure alone.

**Questions:**

-

---

> ### Author Response · Authors · 2024-11-29
>
> Thank you for your thorough analysis and constructive feedback on our paper. We appreciate the opportunity to clarify the points raised and to provide additional insights into our research.
>
> > **Weakness 1.** Experimental validation - while I appreciate the use of real world video the experimental validation is lacking. There are only two tasks used and if a method is aiming to show improvement in continual learning then I would really expect more. For example, including more datasets (Ego4D, SSv2 for example) and more tasks (dense tasks, pixel prediction) would have made the case of the paper stronger.
>
> We currently tackle complex task settings through Kinetics-700 and Epic-Kitchens-100 (EK-100) as also illustrated in the table below. In particular, the EK-100 dataset covers fine-grained tasks with hand-object manipulation posing full / partial occlusions, multi-viewpoints and distribution shifts that were not tackled by earlier works. Nevertheless, we will strive to include more video datasets in our final version.
>
> | Dataset                | Longest Video Length | Average Video Length  | # of Object or Action Categories | Video-understanding Setting    | Used In                     |
> |------------------------|----------------------|-----------------------|------------------|--------------------------|----------------------------|
> | ActivityNet            | 600 (10 mins)       | 120 secs             | 203              | short                    | SMILE [1], vCLIMB [2]        |
> | Kinetics (400/600/700) | 20 secs             | 10 secs              | 400 / 600 / 700  | short                    | SMILE [1], vCLIMB [2], Ours        |
> | UCF101                 | 8 secs              | 5-7 secs             | 101              | short                    | ST-Prompt [3], FrameMaker [4]      |
> | HMDB51                 | 6 secs              | 6 secs               | 51               | short                    | ST-Prompt [3], FrameMaker [4]      |
> | Something-Something V2 | 6 secs              | 4-6 secs             | 174              | short, fine-grained      | ST-Prompt [3], FrameMaker [4]      |
> | Epic-Kitchens-100      | 5400 (1.5 hrs)      | 900-1200 secs (15-20 mins) | 331          | long, fine-grained       | Ours                 |
>
> > **Weakness 2.** Analysis - there is very little analysis as to what the model learns and how - the main ablation is the previous task buffer size, the rest is in the appendix but not a lot of analysis of the significance of the results is given. I would have loved to see how the compressed representation evolve as more tasks are introduced - do they stay the same? do they change abruptly to fit the new task (while still being meaningful for the old ones)? some visualization of the learned representation would be nice as well.
>
> We would like to request for clarification on what kind of experiment would be sufficient to demonstrate this. The representations do change – that is the purpose of the CL procedure, and is quantified by the forgetting metric. We can quantify change in the representation space for example using L2 distances, but being a learned space, distances are difficult to interpret. We would also appreciate suggestions for any specific methods for visualisations.
>
> > **Weakness 3.** Clarity - I found the paper hard to follow. The model and problem set up are not well explained and the figure captions do little to help. Specifically, the method section (4) needs more context with a clear definition of what tasks are and how they evolve over time. Figure 2 caption should be extended - the model is quite simple (I think) and should be completely understandable from that figure alone.
>
> We would like to kindly ask for more details on what parts of the Method section are confusing, in order for us to improve them. As for the definition of tasks (identified with index $t$), they are independent distributions of labels/classes, which are different over time. CL in general is concerned with such evolving (non-I.I.D.) distributions. We have just added a new Figure 2, which we hope clarifies the information flow in our method, compared to other attempts at compressed buffers.
>
> ---
>
> References
>
> 1. SMILE: "Just a Glimpse: Rethinking Temporal Information for Video Continual Learning", CVPR 2023.
> 2. vCLIMB: "A Novel Video Class Incremental Learning Benchmark", CVPR 2022.
> 3. ST-Prompt: "Space-time Prompting for Video Class-incremental Learning", ICCV 2023.
> 4. FrameMaker: "Learning a Condensed Frame for Memory-Efficient Video Class-Incremental Learning.", NeurIPS 2022.
>
> We are grateful for the chance to discuss our work's potential, and wish to thank you again for your valuable input.

---

> > ### Comment · Reviewer_ngdC · 2024-12-03
> > **Thank you for your responses**
> >
> > While I appreciate the time taken by the authors to respond, I don't think the reponses addresses all of my concerns. I am glad that more datasets will be added to a future version of the paper - that would definitely make the case for the paper stronger.
> > However I still think the paper lacks clarity (though the revised manuscript is improved) and analysis.
> >
> > One final note - the responses to the reviews came very late in the discussion period and this does not allow for sufficient time for proper discussion. I am keeping my score as it is and encourage the authors to further refine and improve the work and submit to a future venue.

---

> > > ### Author Response · Authors · 2024-12-04
> > >
> > > We would like to thank the reviewer for the encouragement, but most of all we would appreciate specific pointers to areas that lack clarity, as this would help us improve the paper further. Due to the high volume of points to respond over the 4 reviews, we could not reply any sooner than we did. But specific suggestions for clarity and analysis are still useful, even if we do not get a chance to respond further.

---

### Official Review · Reviewer_h5dS · 2024-11-03

**Soundness:** 3
**Presentation:** 2
**Contribution:** 3
**Rating:** 6
**Confidence:** 3

**Summary:**

This work implements continual learning for action and object classification in relatively long video clips. This is an important setting for many applications such as robotics, and is quite challenging due to the high information density and temporal correlations inherent in video data. The authors employ a VQ-VAE-based video compression approach to enable large-scale storage of encoded video information in a buffer, enabling replay of previously encountered examples to mitigate catastrophic forgetting in incremental learning settings from scratch and with pretraining. The compression strategy is designed to balance stability and plasticity, using a frozen decoder for each task to minimize representational drift. The proposed algorithm outperforms several relevant baselines by large margins under memory-constrained conditions.

**Strengths:**

1. The described setting (continual learning of classification tasks involving long videos as input) is relevant to many practical applications in robotics, security camera systems, and other areas – it is also quite challenging due to the size of video data and the inherent temporal correlations, and as such has been explored by existing work to only a limited extent.

2. Replay-based continual learning methods in image processing applications can have a large memory storage footprint – this is exacerbated with video data, making approaches like this one especially practically useful in this setting.

3. Combining a stored set of frozen “decompressors” to manage representational drift with a “compressor” trained on-the-fly is an interesting and novel approach to this continual learning problem. Figure 2 is well-designed and quite helpful for understanding the approach.

4. The proposed approach outperforms the baselines on all benchmarks, and often by large margins. The selected baselines are appropriate and are compared with the proposed method in reasonable ways.

5. The paper is well-written, and for the most part is clear and easy to follow. For example, the methods section is written in a way that makes the proposed approach easy to understand, by first presenting the simplified IID case and then moving to the incremental learning case. There is an insightful and balanced account of biological inspiration and plausibility of the proposed algorithm in the introduction.

**Weaknesses:**

This paper appears to present strong state-of-the-art results on an important and challenging continual learning problem, but the review score is limited primarily due to insufficient detail in describing and justifying the proposed algorithm and in describing the setting/datasets. Performance comparisons are also not presented in a sufficiently rigorous way (no estimates of uncertainty, no clear definition of the accuracy metric being used). However, the weaknesses of the paper appear relatively addressable in ways that could improve this reader’s review score.

1.	The proposed method uses an existing video compression algorithm to allow a large portion of compressed video data to be stored in a buffer for replay, with novelty mainly arising from the specific configuration of encoders and decoders and how they are trained or kept frozen at different stages of continual learning in different settings (e.g., keeping a separate decompressor for stored codes from each task) – however, this configuration is not strongly justified either theoretically or empirically (see also items 1 and 2 in the “questions” section).

2.	In the related works section under “Continual Learning with Images and Videos”, there is only one reference to an existing work on continual learning with videos. To make the claim that this is the first practical CL algorithm in a large-scale long video setting would seem to require a more thorough review of prior approaches (even if they do not fully meet this criterion) to distinguish the current work from them – for example, the authors could consider the following:
a.	Verwimp, Eli, Kuo Yang, Sarah Parisot, Lanqing Hong, Steven McDonagh, Eduardo Pérez-Pellitero, Matthias De Lange, and Tinne Tuytelaars. "Clad: A realistic continual learning benchmark for autonomous driving." Neural Networks 161 (2023): 659-669.
b.	Wu, Jay Zhangjie, David Junhao Zhang, Wynne Hsu, Mengmi Zhang, and Mike Zheng Shou. "Label-efficient online continual object detection in streaming video." In Proceedings of the IEEE/CVF International Conference on Computer Vision, pp. 19246-19255. 2023.

3.	There is little to orient the unfamiliar reader with the overall setting, specifically the EpicKitchens and Kinetics-700 datasets.  It would be useful to include some additional details, such as basic statistics on how long the videos are, examples of the kinds of actions/objects that are depicted in the datasets, how the labeling works (e.g., does each frame of the video have one label and one label only? How do the models and the labeling schemes manage smooth transitions between classes?) and visualizations of a few examples (there are a few examples from Epic-Kitchen in Figure 1, but none from Kinetics-700 and there is no in-text reference to Figure 1).

4.	Although the proposed method appears to outperform the baselines by large margins, there is no way to assess the statistical reliability of these results. I think it is important to include results from multiple training runs and assess variability among runs (e.g., reporting standard error or confidence intervals) in addition to the mean performance numbers, and to add error bars to Figure 3.

**Questions:**

1.	Additional ablation studies could further justify aspects of the proposed approach. In particular, it would be interesting to explore the benefits and drawbacks of the specific way in which the autoencoder is trained. For the incremental setting, is it necessary to keep a separate decoder for each task to limit representational drift, or does the method perform well using only one decoder that is trained continuously? Does it improve performance to maintain a separate encoder for each task in addition to a separate decoder? Any performance tradeoffs here should be described alongside the drawbacks of maintaining more encoders/decoders – what is the size in memory of the autoencoder parameters relative to the replay buffer? For example, if it so happens that it is very cheap to store lots of different encoders/decoders for each task, this approach might be well-justified if it also improves performance.

2.	Related to the above, it does not seem entirely clear what is meant by the compressor being trained “continuously” in the incremental setting. Is it that there is just one compressor that continues to be updated with each task? Or is a separate compressor trained from a random initialization for each task? Or, at the conclusion of each task, is the compressor for that task frozen and a copy made of it to form the initial condition of the compressor to be trained for the next task?

3.	There are some prior works that have explored continual learning in video-formatted datasets. One claim that underscores the novelty/significance of the work is that these video clips are much longer – can you provide a measure of quantification for this? How much longer, and is this a practically meaningful increase in duration of videos that can be processed?

4.	In section 5.1 where does the 224x224x14 dimension of each video clip come from? Are these grayscale videos with 14 frames? (this would seem inconsistent with the statement that the method operates on long videos)

5.	In the “baselines” section, a limit on the number of samples in the buffer per task is described. However, it is not described how these samples were selected – e.g., perhaps they were randomly (IID) selected from each task, in which case this should be made explicit. There is a statement in section 6.1 that “One interesting finding from our work is that we do not need to apply any frame selection or sampling strategy, even for very large videos,” however it is not clear what this means – is it that the compression is so efficient that you can store every single frame? Or is it that random selection is sufficient? (this strategy is commonly used in replay-based continual learning approaches). For the sampling selection strategies of the baselines, I see that some are explained in the appendix, although it is not clear how the sampling worked for REMIND.

6.	Are the incremental and pretraining settings here best characterized as class-incremental learning or task-incremental learning? (i.e., when the trained model is evaluating a new, unknown sample, does it also need to be told which task the sample belongs to?)

7.	What is meant by “average accuracy” in tables such as table 1? This can be measured in different ways – for example, it could be average accuracy on all tasks measured at the conclusion of the task sequence, or it could also be averaged across accuracy measured after each task increment.

Minor comments:

8.	Some of the references appear to be incorrectly formatted – e.g. [1], [3], [6], and many more do not have a journal or conference listed. A few also have incomplete author information (e.g., [1] does not list an author, only the title and year). It is also my understanding that in-text citations should be author-date formatted instead of just numbers for each reference (specifically for ICLR).

9.	There are a few typos - e.g., in section 3.2 “a concatenation of m samples from each of the task” and near the end of section 5.3 “we store the resulting the codes.” Additional proofreading would be helpful to refine the paper.

10.	The average forgetting (AvgF) metric should be briefly defined in the paper – currently, there is just a citation to the Avalanche GitHub repository.

11.	The method seems to be referred to as “BootstrapCL” in some of the tables, but this name is not introduced anywhere else in the text. Why is it called “BootstrapCL”? It should be made more clear that this is the name of the new algorithm – e.g., in the tables it could be called “BoostrapCL (Ours)”. It can also be helpful to bold the best performance numbers on each metric in the tables.

12.	Equation 10 seems to imply that the same encoder is used for both new samples and samples reconstructed from the buffer. Why is it that the decoder from previous tasks needs to be retained to decode those older examples, but the same encoder can be used for all tasks?  It is seemingly contradictory that, in equations 6 and 7, there appear to be different versions of the encoder for each task ($ϕ_1$, $ϕ_2$, etc.) when it is also stated that the encoder is trained continuously in the incremental setting.

13.	I suggest combining the ablation study tables 3-5 in the appendix into a single table, so it is easier to compare the performance under each ablation with the baseline performance and also compare among the different ablations.

14.	If I understand correctly, “compressor” is used interchangeably with “encoder” and “decompressor” with “decoder.” I suggest choosing one set of terms and using them throughout the paper consistently.

15.	The CL acronym for continual learning should also be used consistently.

---

> ### Author Response · Authors · 2024-11-29
>
> Thank you for your recognition of our work and for your valuable feedback.
>
> > **Weakness 1.** The proposed method uses an existing video compression algorithm to allow a large portion of compressed video data to be stored in a buffer for replay, with novelty mainly arising from the specific configuration of encoders and decoders and how they are trained or kept frozen at different stages of continual learning in different settings (e.g., keeping a separate decompressor for stored codes from each task) – however, this configuration is not strongly justified either theoretically or empirically (see also items 1 and 2 in the “questions” section).
>
> We address this in our responses to question 1 and 2.
>
> > **Weakness 2.** In the related works section under “Continual Learning with Images and Videos”, there is only one reference to an existing work on continual learning with videos. To make the claim that this is the first practical CL algorithm in a large-scale long video setting would seem to require a more thorough review of prior approaches (even if they do not fully meet this criterion) to distinguish the current work from them.
>
> We thank the reviewer for pointing out additional references – we added CLAD and Efficient-CLS to Related Works.
>
> > **Weakness 3.** There is little to orient the unfamiliar reader with the overall setting, specifically the EpicKitchens and Kinetics-700 datasets. It would be useful to include some additional details, such as basic statistics on how long the videos are, examples of the kinds of actions/objects that are depicted in the datasets, how the labeling works (e.g., does each frame of the video have one label and one label only? How do the models and the labeling schemes manage smooth transitions between classes?) and visualizations of a few examples (there are a few examples from Epic-Kitchen in Figure 1, but none from Kinetics-700 and there is no in-text reference to Figure 1).
>
> We are adding the following descriptions to the Appendix (due to lack of space). For convenience, we reproduce them here:
>
> * Epic-Kitchens-100: The average video length is 20 minutes, longest video length is 1.5 hours and shortest video length is 5 minutes. Total video footage length is 100 hours. Each video is at 25 frames per second. We also describe the annotations of the dataset. Each video is associated with a participant and video identifier. Each video is split into a block of frames (segment) with a start and a stop timestamp, and indicated with the start and stop frame. A video segment is labeled with all the noun categories present in it (so multiple labels per clip).
>
> The following are some example annotations:
> | Participant ID | Video ID | Start timestamp | Stop timestamp | Start frame | Stop frame | Nouns                      | Noun classes |
> |----------------|----------|-----------------|----------------|-------------|------------|---------------------------|--------------|
> | P01            | P01_01  | 00:00:29.22     | 00:00:31.32    | 1753        | 1879       | ['fridge']                | [12]         |
> | P01            | P01_01  | 00:09:07.40     | 00:09:09.01    | 32844       | 32940      | ['container', 'fridge']   | [21, 12]     |
> | P01            | P01_105 | 00:00:27.01     | 00:00:27.83    | 1350        | 1391       | ['container', 'cupboard'] | [21, 3]      |
> | P02            | P02_108 | 00:00:43.83     | 00:00:45.92    | 2191        | 2296       | ['biscuit', 'cupboard']   | [104, 3]     |
>
>
> As seen above, the labeling is not at frame-level but instead at the video segment level. There are a total of 331 noun classes covering various nouns involved in kitchen actions (including everyday equipment). Smooth transitions between classes are ensured by presenting the segments to the models chronologically.
>
> * Kinetics-700: The average video length is 10 seconds, longest video length is 15 seconds and shortest video length is 7 seconds. Each video is at 25 frames per second. There are 700 classes in total, and each class label is also associated with an integer label (which is an integer value from 0 to 699). Each video is associated with a class label (and its corresponding integer label).
>
> The following are some example annotations:
> | Label                | Youtube ID   | Start time | Stop time |
> |----------------------|--------------|------------|-----------|
> | "baking cookies"     | JJWwLganiil  | 31         | 41        |
> | "gymnastics tumbling"| 5KbfOS44-gM  | 49         | 59        |
> | "writing"            | iYcARQA6VIU  | 0          | 10        |
> | "wrapping present"   | Qo5lspgmqPU  | 167        | 177       |
>
> We have added some qualitative examples for Kinetics-700 to the Appendix and will add more in the final version similar to Epic Kitchens-100.

---

> ### Author Response · Authors · 2024-11-29
>
> > **Weakness 4.** Although the proposed method appears to outperform the baselines by large margins, there is no way to assess the statistical reliability of these results. I think it is important to include results from multiple training runs and assess variability among runs (e.g., reporting standard error or confidence intervals) in addition to the mean performance numbers, and to add error bars to Figure 3.
>
> We are running multiple seeds, and will add the results in the final version shortly. So far, there doesn't seem to be any significant variability across different runs.
>
>
> > **Q1.** Additional ablation studies could further justify aspects of the proposed approach. In particular, it would be interesting to explore the benefits and drawbacks of the specific way in which the autoencoder is trained. For the incremental setting, is it necessary to keep a separate decoder for each task to limit representational drift, or does the method perform well using only one decoder that is trained continuously? Does it improve performance to maintain a separate encoder for each task in addition to a separate decoder? Any performance tradeoffs here should be described alongside the drawbacks of maintaining more encoders/decoders – what is the size in memory of the autoencoder parameters relative to the replay buffer? For example, if it so happens that it is very cheap to store lots of different encoders/decoders for each task, this approach might be well-justified if it also improves performance.
>
> > **Q2.** Related to the above, it does not seem entirely clear what is meant by the compressor being trained “continuously” in the incremental setting. Is it that there is just one compressor that continues to be updated with each task? Or is a separate compressor trained from a random initialization for each task? Or, at the conclusion of each task, is the compressor for that task frozen and a copy made of it to form the initial condition of the compressor to be trained for the next task?
>
> We jointly address questions 1 and 2 here. We updated the paper with figure 2 and added a table below to address the above questions. Figure 2 illustrates the information flow from the previous decoder, to the current encoder, and finally to the current classifier (last column). We do not retain the decoders from all the previous CL tasks, instead we retain the one from the previous task only. We use the (single) previous decoder to reconstruct all the codes stored in the buffer. At each CL task, we instantiate the current autoencoder with the one from the previous task. Additionally, at the end of each CL task, the current encoder has refreshed all the codes in the buffer (i.e. the codes were updated to work with the current decoder, instead of the previous one), and as a result we can use the same encoder for all the tasks.
>
> Figure 2 (column 3) shows that while keeping separate autoencoders for each past task would not result in representational drift, it would lead to an unbounded memory budget that scales with the number of tasks. Relative to this, our proposed scheme (column 4) refreshes codes to keep them from drifting, while only requiring a single snapshot of the last decoder. The extra total memory budget for the past and current autoencoder is 750 Mb. We also updated Table 2 to add relative memory budgets between buffer and model storage.
>
>
> |                     | Naive SGD (A) | Keep all tasks' AEs (B) | Ours (C) |
> |---------------------|---------------|-------------------------|----------|
> | Memory (#models)    | 1             | N (N=tasks)            | 2        |
> | Representation drift| Yes           | No                     | No       |

---

> ### Author Response · Authors · 2024-11-29
>
> > **Q3.** There are some prior works that have explored continual learning in video-formatted datasets. One claim that underscores the novelty/significance of the work is that these video clips are much longer – can you provide a measure of quantification for this? How much longer, and is this a practically meaningful increase in duration of videos that can be processed?
>
> We process 10x longer video lengths over most prior works. Following table describes each video dataset with the length of its longest video (column 2), average length (column 3), classification and temporal complexity in its video understanding setting (column 4, 5), and the respective CL works these datasets are used in (column 6). By extending to a complex long video setting (such as Epic-Kitchens-100), our method shows a meaningful increase in both video length and complexity of video understanding settings absent in the previous works (as illustrated in the table below).
>
>
> | Dataset                | Longest Video Length | Average Video Length  | # of Object or Action Categories | Video-understanding Setting    | Used In                     |
> |------------------------|----------------------|-----------------------|------------------|--------------------------|----------------------------|
> | ActivityNet            | 600 (10 mins)       | 120 secs             | 203              | short                    | SMILE [1], vCLIMB [2]        |
> | Kinetics (400/600/700) | 20 secs             | 10 secs              | 400 / 600 / 700  | short                    | SMILE [1], vCLIMB [2], Ours        |
> | UCF101                 | 8 secs              | 5-7 secs             | 101              | short                    | ST-Prompt [4], FrameMaker [5]      |
> | HMDB51                 | 6 secs              | 6 secs               | 51               | short                    | ST-Prompt [4], FrameMaker [5]      |
> | Something-Something V2 | 6 secs              | 4-6 secs             | 174              | short, fine-grained      | ST-Prompt [4], FrameMaker [5]      |
> | Epic-Kitchens-100      | 5400 (1.5 hrs)      | 900-1200 secs (15-20 mins) | 331          | long, fine-grained       | Ours                 |
>
>
> > **Q4.** In section 5.1 where does the 224x224x14 dimension of each video clip come from? Are these grayscale videos with 14 frames? (this would seem inconsistent with the statement that the method operates on long videos)
>
> We apologize, but this is the result of a typo in the paper – thank you for pointing it out. We use clips of sizes 224x224x3x32 (32 RGB frames, not grayscale), and in 5.1 we just meant to illustrate the storage size. This is the size of each clip that we encode into a code, and each long video is composed of many such clips / codes. It is also common practice to split a long video to short clips before processing [3]. We will update section 5.1 with this clarification.
>
> > **Q5.** In the “baselines” section, a limit on the number of samples in the buffer per task is described. However, it is not described how these samples were selected – e.g., perhaps they were randomly (IID) selected from each task, in which case this should be made explicit. There is a statement in section 6.1 that “One interesting finding from our work is that we do not need to apply any frame selection or sampling strategy, even for very large videos,” however it is not clear what this means – is it that the compression is so efficient that you can store every single frame? Or is it that random selection is sufficient? (this strategy is commonly used in replay-based continual learning approaches). For the sampling selection strategies of the baselines, I see that some are explained in the appendix, although it is not clear how the sampling worked for REMIND.
>
> Yes, the compression strategy is very efficient, thus it enables our method to store every single frame. This is unlike previous methods, which needed to select frames, due to high storage requirements. A random selection strategy was used for REMIND. We will update the Appendix with this detail.
>
> > **Q6.** Are the incremental and pretraining settings here best characterized as class-incremental learning or task-incremental learning? (i.e., when the trained model is evaluating a new, unknown sample, does it also need to be told which task the sample belongs to?)
>
> This is a class-incremental setting. We do distinguish between training from scratch incrementally (sec. 4.2) and with pre-training (sec. 4.3).
>
> > **Q7.** What is meant by “average accuracy” in tables such as table 1? This can be measured in different ways – for example, it could be average accuracy on all tasks measured at the conclusion of the task sequence, or it could also be averaged across accuracy measured after each task increment.
>
> Yes, it is the average accuracy on all tasks measured at the conclusion of the task sequence.

---

> ### Author Response · Authors · 2024-11-29
>
> > **Comment 1.** Some of the references appear to be incorrectly formatted – e.g. [1], [3], [6], and many more do not have a journal or conference listed. A few also have incomplete author information (e.g., [1] does not list an author, only the title and year). It is also my understanding that in-text citations should be author-date formatted instead of just numbers for each reference (specifically for ICLR).
>
> We thank the reviewer for pointing this out and will fix the citations.
>
> > **Comment 2.** There are a few typos - e.g., in section 3.2 “a concatenation of m samples from each of the task” and near the end of section 5.3 “we store the resulting the codes.” Additional proofreading would be helpful to refine the paper.
>
> We thank the reviewer for pointing this out, and have fixed these typos in the paper.
>
> > **Comment 3.** The average forgetting (AvgF) metric should be briefly defined in the paper – currently, there is just a citation to the Avalanche GitHub repository.
>
> Let $a_{i,t}$  be accuracy on task $i$ of the model that was trained on t tasks, where $i < t$. Average forgetting measures how much performance has degraded across the first $t-1$ tasks. To do so, this metric uses the difference between  best-obtained performance of the desired task and the performance obtained from the current incremental learner.
>
> \begin{equation}
> F_t = \frac{1}{t-1} \sum_{1}^{t-1} f_{i,t} \quad \text{where} \quad f_{i,t} = \max_{q<t} \left( a_{i,q} - a_{i,t} \right)
> \quad \text{or} \quad f_{i,t} = a_{i,i} - a_{i,t}
> \end{equation}
>
> We thank the reviewer for pointing this out and updated the Appendix with this definition.
>
> > **Comment 4.** The method seems to be referred to as “BootstrapCL” in some of the tables, but this name is not introduced anywhere else in the text. Why is it called “BootstrapCL”? It should be made more clear that this is the name of the new algorithm – e.g., in the tables it could be called “BoostrapCL (Ours)”. It can also be helpful to bold the best performance numbers on each metric in the tables.
>
> The name is a reference to the fact that our CL method bootstraps each compressor from the previous one – we will make this more clear in the paper. We also added “BoostrapCL (Ours)” to the tables.
>
> > **Comment 5.** Equation 10 seems to imply that the same encoder is used for both new samples and samples reconstructed from the buffer. Why is it that the decoder from previous tasks needs to be retained to decode those older examples, but the same encoder can be used for all tasks? It is seemingly contradictory that, in equations 6 and 7, there appear to be different versions of the encoder for each task (, , etc.) when it is also stated that the encoder is trained continuously in the incremental setting.
>
> We understand that a source of confusion might have been the omission (for ease of notation) of a subscript for the encoder in Eq. 10. We added back the subscript to make it clear that the encoder is the one for the current task. There is no contradiction if one understands our method as a sequence of optimization problems (eq. 7 and 10), first optimized with samples from task 1, then with those from task 2, and so on. When optimizing the encoder/decoder/classifier for task $t$, the decoder for task $t-1$ is constant/frozen (and any previous ones are not used at all).
>
> To make this clear, we would like to direct the reviewer’s attention to Fig. 2 (added to the paper and as described in question 1 and 2), which illustrates the information flow from the previous decoder, to the current encoder, and finally to the current classifier (last column).
>
> In summary, we do not retain the decoders from all the previous CL tasks, instead we retain the one from the (single) previous task only. We use the previous decoder to reconstruct all the codes stored in the buffer.  At each CL task, the latest encoder refreshes all the codes in the buffer, as a result we can use the same encoder for all the tasks in Equation 10.
>
> Equation 10 represents the classification objective which is applied per CL task. It is applied after the code refreshment, so it uses the latest encoder.
>
> > **Comment 6.** I suggest combining the ablation study tables 3-5 in the appendix into a single table, so it is easier to compare the performance under each ablation with the baseline performance and also compare among the different ablations.
>
> We thank the reviewer for the suggestion, and will combine the ablation study into a single table.

---

> ### Author Response · Authors · 2024-11-29
>
> > **Comment 7.** If I understand correctly, “compressor” is used interchangeably with “encoder” and “decompressor” with “decoder.” I suggest choosing one set of terms and using them throughout the paper consistently.
>
> We thank the reviewer for the suggestion, and will update the paper with using one set of terms.
>
> > **Comment 8.** The CL acronym for continual learning should also be used consistently.
>
> We use continual learning (CL) in section titles (to be self-contained) and to define CL at the start of major sections, while using the initialism CL elsewhere, which we believe is consistent. We would also appreciate suggestions on how to best use it.
>
>
> ---
>
> References:
>
> 1. SMILE: "Just a Glimpse: Rethinking Temporal Information for Video Continual Learning", CVPR 2023.
> 2. vCLIMB: "A Novel Video Class Incremental Learning Benchmark", CVPR 2022.
> 3. TQN: "Temporal Query Networks for Fine-grained Video Understanding", CVPR 2021.
> 4. ST-Prompt: "Space-time Prompting for Video Class-incremental Learning", ICCV 2023.
> 5. FrameMaker: "Learning a Condensed Frame for Memory-Efficient Video Class-Incremental Learning.", NeurIPS 2022.
>
>
> We are grateful for the chance to discuss our work's potential, and wish to thank you again for your valuable input.

---

> > ### Comment · Reviewer_h5dS · 2024-12-03
> > **Response to rebuttal**
> >
> > My thanks to the authors for their detailed responses to my review. I am providing some follow-up responses below.
> >
> > **Referring to Weakness 1 (and questions 1 and 2) from the original review:**
> >
> > The newly added Figure 2 (in revised version) is very helpful for understanding the proposed method - I had previously not realized that the stored codes are ``refreshed'' during each task by decoding them with the older decoder and then re-encoding them with the new task's encoder - this strategy makes sense because it limits both representational drift and the overhead of storing many decoders in memory.
> >
> >
> > **Referring to Weakness 3 from the original review:**
> >
> > The additions to the appendix describing the datasets are very helpful. I understand that there is limited space, but for the sake of the reader being better able to follow the methodology I still suggest squeezing in at least a 1-2 sentence summary of the format of the datasets in the methods section.
> >
> >
> > **Referring to Weakness 4 from the original review:**
> >
> > It is encouraging to hear that the authors have initiated runs with multiple random seeds to assess variability. In the current manuscript, I still do not see any uncertainty estimates - although I understand that completing multiple runs can take time. In my view, uncertainty estimates/error bars would need to be added before publication, both for the proposed method and baselines (e.g., in Figure 4 of the revised manuscript).
> >
> >
> > **Referring to Question 3 from the original review:**
> >
> > This new Table is helpful for clarifying the large jump in video length addressed in this paper compared with prior works. I suggest sorting the rows of the Table by average video length.
> >
> >
> > **Referring to Question 5 from the original review:**
> >
> > Thank you for this clarification. I suggest stating explicitly in the main text that the proposed method stores every single frame - it is otherwise not obvious that this should be the case, especially with memory-intensive video data.
> >
> > **Referring to Question 6 from the original review:**
> >
> > Thank you for this clarification also, that the proposed method works in a class-incremental setting (i.e., task identity is not required during inference). This is probably also worth stating explicitly somewhere in the paper.
> >
> > **Referring to Comment 6 from the original review:**
> >
> > It is good to hear that the authors plan to combine the ablation study into a single table. This is just a reminder to please remember to do this for the final version, as they are still separate tables in the current revision.
> >
> > **Summary**
> >
> > Overall, my concerns have mostly been addressed (or are in the process of being addressed, i.e. multiple runs for uncertainty estimates) and the manuscript has been improved particularly with the addition of Figure 2 - I am raising my score to ``marginally above the acceptance threshold.'' I think that the paper still requires substantial revisions to improve its clarity before publication (some of which are noted above).

---

> ### Author Response · Authors · 2024-12-04
>
> We are glad that the latest modifications indeed improve the paper’s clarity. We thank the reviewer for the additional suggestions, and will be sure to include them in the next version. We would like to stress that, although re-running with additional seeds is time-consuming, with the ones we ran so far we did not see any deviations from the trends already shown in the experiments, and we expect the full set of replications to not change the conclusions. We will be sure to include the full set of uncertainty estimates / error bars in the revised manuscript both for the proposed method and baselines (in the experiment tables and Figure 4).

---

### Official Review · Reviewer_nQAS · 2024-11-04

**Soundness:** 2
**Presentation:** 3
**Contribution:** 2
**Rating:** 3
**Confidence:** 5

**Summary:**

The paper presents a memory-efficient approach for video continual learning (CL) using compressed embeddings stored in a neural-code rehearsal buffer. The main idea is to reduce the high memory demands of video CL by compressing video frames into compact neural codes instead of storing raw data. The method also includes a code-refreshing mechanism to mitigate representational drift, which may happen as the model continues the incremental learning process. The method is evaluated on Epic-Kitchens-100 and Kinetics-700, across both pre-trained and completely incremental learning settings. Empirical results indicate that the method achieves promising performance with significantly reduced memory usage.

**Strengths:**

1. **Reasonable Approach to Memory Efficiency**: The paper introduces a novel memory-efficient method for video continual learning by storing compressed neural codes rather than raw frames. This approach, combined with a code-refreshing mechanism, is a reasonable way to adapt continual learning to video data’s storage constraints and combat representational drift and catastrophic forgetting.

2. **Clear Experimental Setup**: The experiments are well-structured, covering both pre-training and incremental learning settings on widely-used large-scale video datasets (Epic-Kitchens-100 and Kinetics-700). Memory constraints and compression rates are clearly defined.

3. **Potential Significance for Real-World Applications**: By focusing on reducing memory demands in video CL, the paper tackles a central obstacle in scaling continual learning to real-world applications. This approach could be impactful for memory-limited devices and applications requiring continual processing of video data, such as surveillance or autonomous systems.

**Weaknesses:**

1. **Limited Novelty in Memory Efficiency Solutions**
   While the paper proposes a new method to address memory efficiency in video CL, this problem has already been identified and approached by prior works. From a benchmarking perspective, **vCLIMB** [1] redefined the memory metric specifically for video CL, proposing **Memory Frame Capacity** to measure memory usage in terms of frames rather than full video instances. This framework allows for evaluating frame selection strategies in video CL. From a method perspective, Furthermore, vCLIMB implemented a regularization term to reduce representation drift between original videos and stored frames, improving memory efficiency in rehearsal-based CL. Additionally, **FrameMaker** [2] further addresses memory efficiency by introducing **Frame Condensing**, where a single condensed frame per video is stored along with instance-specific prompts to retain temporal details. By not comparing against these methods, the paper’s memory efficiency claim is weakened, as the approach lacks context relative to prior works.

2. **Lack of Comparison to Rehearsal-Free Methods**
   If memory efficiency is a primary goal, comparisons with **rehearsal-free video CL methods** are essential, as these approaches inherently avoid memory constraints. For instance, **ST-Prompt** [3] achieves continual learning without rehearsal by using vision-language models and temporal prompts to encode sequential information, thus sidestepping the need for a memory buffer. More recently, **DPAT (Decoupled Prompt-Adapter Tuning)** [4] combines adapters for capturing spatio-temporal information with learnable prompts, employing a decoupled training strategy to mitigate forgetting without rehearsal. While DPAT may be too recent for comprehensive testing, at minimum, a comparison to ST-Prompt or a discussion on why rehearsal-free methods were not included would provide a more complete assessment of memory efficiency in CL.

3. **Inadequate Baselines for Modern CL Standards**
   The paper’s use of **GDumb** [5] as a baseline is insufficient for evaluating the performance of a modern CL method. GDumb, introduced in 2020, was meant to highlight flaws in existing CL evaluation metrics and methods, demonstrating that a simple random-sampling rehearsal method could outperform many complex algorithms of that time. However, it is not representative of state-of-the-art continual learning. Since its release, more advanced rehearsal-based methods, such as **ER-ACE** [6] and **L2P** [7] have been developed, each addressing the limitations GDumb originally exposed. GDumb’s rudimentary approach lacks the complexity needed to benchmark against a method claiming novel contributions in memory-efficient CL, and thus relying on GDumb alone creates an unconvincing evaluation framework for the proposed method. Including state-of-the-art baselines from both image and video CL (see previous point for video baselines) would strengthen the paper’s claims of memory efficiency and performance.

4. **Insufficient Justification of Benchmark Superiority**
   The paper introduces a new benchmark with a pre-training phase on a subset of classes, followed by incremental learning. However, **Park et al. (2021)** [8] has already explored a similar pre-training and incremental learning setup for video CL. The paper does not provide sufficient justification for why its benchmark is necessary or superior to existing benchmarks (such as [1] and [8]). A new benchmark should ideally improve upon current setups in aspects such as realism, task granularity, or sequence transitions. Without a clear rationale, the proposed benchmark appears redundant rather than an improvement.

5. **Unsubstantiated Novelty Claim in Large-Scale, Long-Video Testing**
   The paper claims to be the first to extend CL to “large-scale naturally-collected long videos.” This claim is inaccurate, as several previous studies have conducted video CL on large, untrimmed datasets. For example, **vCLIMB** and other works used **ActivityNet** [1] for CL, which includes long, untrimmed videos from natural events and provides extensive temporal context. Similarly, the **Kinetics** and **Something-Something** datasets have been widely used for video CL research, with recent methods like **DPAT** [4] even leveraging Epic-Kitchens for long, naturally collected video scenarios. Without clear evidence that the benchmark adds unique value, such as in video length or task diversity, the claim of novelty is misleading and diminishes the contribution’s significance.

---

### References:
1. vCLIMB: "A Novel Video Class Incremental Learning Benchmark", CVPR 2022.
2. FrameMaker: "Learning a Condensed Frame for Memory-Efficient Video Class-Incremental Learnin.", NeurIPS 2022.
3. ST-Prompt: "Space-time Prompting for Video Class-incremental Learning", ICCV 2023
4. DPAT: "Decoupled Prompt-Adapter Tuning for Continual Activity Recognition", CoLLAs 2024.
5. GDumb: "A Simple Approach That Questions Our Progress in Continual Learning", ECCV 2020.
6. ER-ACE: “New Insights on Reducing Abrupt Representation Change in Online Continual Learning”, ICLR 2022.
7. L2P: “Learning to Prompt for Continual Learning”, CVPR 2022.
8. Park et al. (2021): "Class-Incremental Learning for Action Recognition in Videos", ICCV 2021.

**Questions:**

1. **Comparison to Advanced Video CL Methods**: How does the proposed method compare with other recent memory-efficient video CL approaches like vCLIMB and FrameMaker, which use selective frame retention with temporal consistency regularization and condensed frames? These comparisons could contextualize the memory benefits claimed in the paper.

2. **Evaluation Against Rehearsal-Free Methods**: Since memory efficiency is a key focus, why were rehearsal-free methods like ST-Prompt not included as baselines? Including or discussing these could provide a clearer assessment of the method’s memory advantages.

3. **Justification of Benchmark Novelty**: The paper introduces a new benchmark setup with pre-training followed by incremental learning. Could the authors elaborate on why this setup is preferable or unique compared to existing video CL benchmarks? Quantifying the differences and summarizing them in a table might be useful here.

4. **Rationale behind Baselines**: Could the authors explain why the baselines were chosen, including GDumb?

5. **Clarification of “Large-Scale, Naturally-Collected, Long Videos” Claim**: The paper claims to be the first to use “large-scale, naturally-collected long videos” in CL, but prior works have used datasets like ActivityNet, Kinetics, and Something-Something. Could the authors clarify what sets this benchmark apart from these established datasets?

---

> ### Author Response · Authors · 2024-11-29
>
> Thank you for your thorough analysis and constructive feedback on our paper. We appreciate the opportunity to clarify the points raised and to provide additional insights into our research.
>
> > **Weakness 1.** Limited Novelty in Memory Efficiency Solutions
> While the paper proposes a new method to address memory efficiency in video CL, this problem has already been identified and approached by prior works. From a benchmarking perspective, vCLIMB [1] redefined the memory metric specifically for video CL, proposing Memory Frame Capacity to measure memory usage in terms of frames rather than full video instances. This framework allows for evaluating frame selection strategies in video CL. From a method perspective, Furthermore, vCLIMB implemented a regularization term to reduce representation drift between original videos and stored frames, improving memory efficiency in rehearsal-based CL. Additionally, FrameMaker [2] further addresses memory efficiency by introducing Frame Condensing, where a single condensed frame per video is stored along with instance-specific prompts to retain temporal details. By not comparing against these methods, the paper’s memory efficiency claim is weakened, as the approach lacks context relative to prior works.
>
>
> > **Q1.** Comparison to Advanced Video CL Methods: How does the proposed method compare with other recent memory-efficient video CL approaches like vCLIMB and FrameMaker, which use selective frame retention with temporal consistency regularization and condensed frames? These comparisons could contextualize the memory benefits claimed in the paper.
>
>
> Currently, we have a baseline comparison with SMILE [5] which is a more recent work from the authors of vCLIMB [3], outperforming their method in vCLIMB. SMILE has a 2-4 times lower memory budget [5] and shows 2.87% accuracy gains in ActivityNet and 20.2% accuracy gains in Kinetics-400 over vCLIMB [5]. We have added baseline comparisons with vCLIMB [3] in Table 1. Further, ours is a representation learning method, and FrameMaker [4] starts with learned representations due to ImageNet initialization to all the backbone networks in their method. Additionally, there is significant overlap between Imagenet classes and Kinetics-700 classes, which results in a different setting than incremental learning from scratch, or from a limited (non-overlapping) set of classes (which are what we use in experiments).
>
> We also address concerns regarding Memory Frame Capacity here; as mentioned under weakness 1, this is a metric that allows for evaluating frame selection strategies in video CL [3]. In this work, we propose a method that circumvents the need for frame selection in video CL. Due to this, the metric does not fit our scenario. Regardless of the number of frames stored, each method has a total memory footprint which can be informative to determine its memory efficiency as seen in Figure 4. Further, we have added vCLIMB to Table 1 in experiments, and will add it to Figure 4 in the final version.
>
> Our work extends evaluation to long video settings. In this setting, research shows that frame selection or condensing are detrimental for video understanding performance [1, 2]. This is due to the loss in temporal resolution and continuity crucial for fine-grained and long-term context preservation as described in Prince and Damen (2019) [1] and TIM [2]. In contrast, vCLIMB [3], FrameMaker [4] and SMILE [5] report performance in short-video settings where detrimental effects from frame selection or condensing is negligible due to simpler temporal dynamics. Therefore, a direct comparison with vCLIMB [3] or FrameMaker [4] with such short-term metrics is not meaningful. A tabular comparison on video datasets used in each of these works is also described in response to question 5, and added to the Appendix.
>
> > **Q2.** Evaluation Against Rehearsal-Free Methods: Since memory efficiency is a key focus, why were rehearsal-free methods like ST-Prompt not included as baselines? Including or discussing these could provide a clearer assessment of the method’s memory advantages.
>
> Rehearsal-free methods like L2P [14] (as mentioned under weakness 3), ST-Prompt [6] and DPAT [7] (as mentioned under weakness 2) rely on large-scale-pre-trained architectures (eg: ImageNet-VIT-B/16, CLIP-VIT-B/16) which can consume up to several hundred gigabytes. This dependency limits the practical usage of these methods in continual learning scenarios where memory resources pose a major bottleneck (such as in edge-based computing - AR, IoT, healthcare). In contrast, our proposed method is significantly lightweight (as also seen in Figure 4 and Table 2), and does not rely on any large-scale pre-trained architecture. Further, with energy, privacy and policy considerations, alternative solutions to large pre-trained architectures may be desirable [8]. We will add discussion on rehearsal-free methods to the related works section.

---

> ### Author Response · Authors · 2024-11-29
>
> > **Q3.** Justification of Benchmark Novelty: The paper introduces a new benchmark setup with pre-training followed by incremental learning. Could the authors elaborate on why this setup is preferable or unique compared to existing video CL benchmarks? Quantifying the differences and summarizing them in a table might be useful here.
>
> Our benchmark setup as described under section 4.3 and under experiments section 5.2 is the same as described in PODNet [10] and Hou et al [12]. So, we would like to clarify that we do not introduce a new benchmark setup, instead mimic the setting as described in PODNet [10] in section 4 under “Experiments” with sub-section “Protocol” or in Hou et al [12] in section 4 under “Experiments”. We have added this clarification to section 4.3 and 5.2 of the paper.
> Under section 4.2 “Evaluation Protocol”, Park et al. (2021) [9] also references PODNet [10] and Hou et al [12] for this benchmark setup.  This setting has several advantages, and those have already been described in earlier works [9, 10, 12]. Further, we have included Park et al. (2021) [9] in the tabular comparison shared in response to question 5.
>
>
> > **Q4.** Rationale behind Baselines: Could the authors explain why the baselines were chosen, including GDumb?
>
> REMIND [15] is a compression-based memory CL method. It is related to our method in that it focuses on compressing the raw RGB input using quantization and storing the resulting compressed codes instead of the RGB input in the replay buffer. It also follows the benchmark setup described in question 3. SMILE [5], as described in question 1, proposed a video CL method relevant for comparison as we also propose a video CL approach. GDumb [11], while a traditional CL work, had a simple implementation and served as a robust evaluation technique. Unlike all prior works, we extend it to a new video CL setting (as described in question 5), so this simple and robust technique served as a sanity check to ensure that we can genuinely outperform naive CL strategies.
> In addition to these, we have added the baseline discussed in response to question 1 (in Table 1 under experiments), and will add more modern CL baselines in the final version.

---

> ### Author Response · Authors · 2024-11-29
>
> > **Q5.** Clarification of “Large-Scale, Naturally-Collected, Long Videos” Claim: The paper claims to be the first to use “large-scale, naturally-collected long videos” in CL, but prior works have used datasets like ActivityNet, Kinetics, and Something-Something. Could the authors clarify what sets this benchmark apart from these established datasets?
>
> Our claim is supported by the qualitative increase in video length compared to these previous works. The following table describes each video dataset with the length of its longest video (column 2), average length (column 3), classification and temporal complexity in its video understanding setting (column 4, 5), and the respective CL works these datasets are used in (column 6). By extending to a large-scale long video setting (such as Epic-Kitchens-100), our method shows a meaningful increase in both video length and complexity of video understanding settings absent in the previous works (as illustrated in the table below).
>
> We thank the reviewer for pointing out that DPAT [7] also has experimental results on a large-scale long-video setting. However, DPAT was published after our submission or was concurrent with it. Furthermore, we describe the limitations associated with DPAT [7] in response to question 2.
>
>
> | Dataset                | Longest Video Length | Average Video Length  | # of Object or Action Categories | Video-understanding Setting    | Used In                     |
> |------------------------|----------------------|-----------------------|------------------|--------------------------|----------------------------|
> | ActivityNet            | 600 (10 mins)       | 120 secs             | 203              | short                    | SMILE [5], vCLIMB [3], DPAT [7]        |
> | Kinetics (400/600/700) | 20 secs             | 10 secs              | 400 / 600 / 700  | short                    | SMILE [5], vCLIMB [3], Ours        |
> | UCF101                 | 8 secs              | 5-7 secs             | 101              | short                    | ST-Prompt [6], FrameMaker [4], Park et al. (2021) [9]      |
> | HMDB51                 | 6 secs              | 6 secs               | 51               | short                    | ST-Prompt [6], FrameMaker [4], Park et al. (2021) [9]      |
> | Something-Something V2 | 6 secs              | 4-6 secs             | 174              | short, fine-grained      | FrameMaker [4], ST-Prompt [6]      |
> | Epic-Kitchens-100      | 5400 (1.5 hrs)      | 900-1200 secs (15-20 mins) | 331          | long, fine-grained       | DPAT [7] (concurrent work), Ours                 |
>
>
>
> ---
>
>
> References
> 1. Prince and Damen (2019): "An Evaluation of Action Recognition Models on EPIC-Kitchens", arXiv preprint arXiv:1908.00867 (2019).
> 2. TIM: "A Time Interval Machine for Audio-Visual Action Recognition", CVPR, 2024.
> 3. vCLIMB: "A Novel Video Class Incremental Learning Benchmark", CVPR 2022.
> 4. FrameMaker: "Learning a Condensed Frame for Memory-Efficient Video Class-Incremental Learning.", NeurIPS 2022.
> 5. SMILE: "Just a Glimpse: Rethinking Temporal Information for Video Continual Learning", CVPR 2023.
> 6. ST-Prompt: "Space-time Prompting for Video Class-incremental Learning", ICCV 2023.
> 7. DPAT: "Decoupled Prompt-Adapter Tuning for Continual Activity Recognition", CoLLAs 2024.
> 8. Strubell et al (2019): Energy and policy considerations for deep learning in nlp, arXiv preprint arXiv:1906.02243 (2019).
> 9. Park et al. (2021): "Class-Incremental Learning for Action Recognition in Videos", ICCV 2021.
> 10. PODNet: "Pooled Outputs Distillation for Small-Tasks Incremental Learning", ECCV 2020.
> 11. GDumb: "A Simple Approach That Questions Our Progress in Continual Learning", ECCV 2020.
> 12. Hou et al. (2019): "Learning a Unified Classifier Incrementally via Rebalancing", CVPR 2019.
> 13. ER-ACE: “New Insights on Reducing Abrupt Representation Change in Online Continual Learning”, ICLR 2022.
> 14. L2P: “Learning to Prompt for Continual Learning”, CVPR 2022.
> 15. REMIND: "REMIND Your Neural Network to Prevent Catastrophic Forgetting", ECCV 2020.
>
> We hope this response has addressed your concerns effectively. We are grateful for the chance to discuss our work's potential, and wish to thank you again for your valuable input.

---

### Official Review · Reviewer_UgR4 · 2024-11-06

**Soundness:** 3
**Presentation:** 3
**Contribution:** 3
**Rating:** 6
**Confidence:** 3

**Summary:**

In this work, the authors propose a method for large-scale long video continual learning to learn from continuous streams without access to the entire dataset. They employ a rehearsal-based approach which reinforces past samples in a memory buffer. To deal with long-videos and continuous streams, they propose to use video codes (video embeddings) instead of raw inputs, and train a video classifier by IID sampling from this buffer.

A video compressor is used to generate the video codes. To deal with the video compressor's catastrophic forgetting, the authors propose continuous compression and decompression technique over the neural-code rehearsal buffer (past video codes). They also train a classifer in the compressed space.

The authors show results on EpicKitchens-100 and Kinetics-700 datasets in two settings --
- (i) incremental learning from scratch, and
- (ii) pretraining.

**Strengths:**

The problem statement is interesting -- continual learning of large-scale long videos from continous video streams.

The proposed technique is reasonable, paper is well-written, and nicely motivated.

The design of the experiments is clearly explained and exhaustive--
- (i) default IID sampling,
- (ii) incremental learning, and
- (iii) CL with pretraining.

For both the incremental learning and CL with pretraining settings, evaluations are done on two large-scale long-video benchmarks -- Kinetics-700 and EpicKitchen-100. The proposed method outperforms the baselines.

**Weaknesses:**

- During the incremental learning stage, the codes in the buffer are decoded using the decoder from the previous task. Can the authors quantify the additional memory required to store decoder weights from the previous task, and compare it with the memory savings from using compressed codes instead of the raw video frames. This would give a clear picture of the overall memory trade-offs in the proposed method.

- Is a single latent code enough to compress/represent a temporally-long and possibly diverse video? Can the authors provide analysis or ablations showing how the performance varies with varying video lengths or video diversity? For instance, can you compare the performance on short vs long videos, or videos with varying amount of scene/action changes.

**Questions:**

What was the number of frames in the videos that were used for training/evaluation? Could you clarify how the performance varies with video length, and whether there's a maximum video length beyond which the method's performance degrades significantly? This would help the readers understand the practical limitations of this approach?

---

> ### Author Response · Authors · 2024-11-29
>
> Thank you for your thoughtful review and for recognizing the importance of our work. We address weaknesses and questions below in two separate comments.
>
> > **Weakness 1.** During the incremental learning stage, the codes in the buffer are decoded using the decoder from the previous task. Can the authors quantify the additional memory required to store decoder weights from the previous task, and compare it with the memory savings from using compressed codes instead of the raw video frames. This would give a clear picture of the overall memory trade-offs in the proposed method.
>
> Our method uses constant additional memory to store the autoencoder weights. We only store the autoencoder from the immediately-previous task, and the current task. For storing both autoencoders, our method uses an additional total memory of 750 Mb. We have also added this additional storage cost in Table 2.
>
> > **Weakness 2.** Is a single latent code enough to compress/represent a temporally-long and possibly diverse video?
>
> We must clarify that one code does not encode a whole video, but rather only a few (32) frames at a time. Each video is split into small blocks of 32 frames (unless otherwise mentioned), and then compressed. That is, every block of frames within the video is compressed independently, instead of the entire video with one code. So, the number of codes varies depending on the video length. A temporally-long video has a higher number of latent codes for it in comparison to a short video.
>
> For Kinetics-700, a video ( with 250 frames on an average) has approximately 8 codes associated with it whereas in Epic Kitchen-100, a video (with 27K frames on an average) has approximately 850 codes associated with it.
>
> > **Weakness 2.** Can the authors provide analysis or ablations showing how the performance varies with varying video lengths or video diversity? For instance, can you compare the performance on short vs long videos, or videos with varying amounts of scene/action changes.
>
>
> * *Analysis on Video Length:* In Epic Kitchens-100 (EK-100), the video length varies from 5 minutes to about 1.5 hours. And, in Kinetics-700 (K-700), the video lengths vary from 7 to 20 seconds. We added tables for our method’s performance based on varying video lengths in response to question 2.
>
> * *Analysis on Video Diversity:* EK-100 and K-700 cover a wide diversity in the videos both within and across the continual learning tasks.
> In K-700, video diversity comes from environmental context changes (eg: swimming / water, skiing / snow), range of motion and tools (e.g., paddleboarding vs. birdwatching), gestures (eg: teaching in a class vs poses during dancing), to name a few. In addition, for each action category in the dataset, the scene and protagonists vary.
> In EK-100, video diversity within each task comes from the same participant shooting at various day times in their kitchen, functionally repurposing various objects, variable scene length and shot type (based on the action performed), objects under multi-viewpoints, partial or full occlusion when captured temporally. In EK-100, video diversity across tasks comes from new and culturally-diverse participants in their respective kitchens and cities. This leads to environmental, cinematography changes and intra-category variations for new or previously-seen objects and actions.
>
> So, both short and long videos, and varying amounts of action and scene changes are covered in each dataset. Further, we ensured that our model is presented with gradual complexity within and across the tasks ensuring smooth transitions (and by presenting data chronologically, wherever applicable) while preserving the diversity.

---

> ### Author Response · Authors · 2024-11-29
>
> > **Q1.** What was the number of frames in the videos that were used for training / evaluation?
>
> For Kinetics-700, we have approximately 14.6 million frames during training and 3 million frames during evaluation. For EK-100, we use 16 million frames during training and 4 million frames during evaluation.
>
> > **Q1.** Could you clarify how the performance varies with video length, and whether there's a maximum video length beyond which the method's performance degrades significantly? This would help the readers understand the practical limitations of this approach?
>
> Following table shows how the performance varies with video length on Epic-Kitchens-100 videos in the Pre-training setting. Specifically, by 2nd task ~6 hours of video length is processed, by 5th task ~15 hours, and by 10th task ~25 hours. Each video is at 25 frames per second. As seen below, the method’s performance remains consistent with increasing video length.
>
> Continual Learning with Pre-training Setting (described in 5.3): Average training (Train) and evaluation (Eval) accuracy at the end of task T on Epic Kitchens-100.
> | Setting                | Task  | 2   | 5   | 10   |
> |------------------------|-------|------|------|------|
> | Pretraining          | Train.| 36.9 | 34.1 | 38.9 |
> |                             | Eval. | 31.2 | 29.8 | 34.8 |
>
> Following table shows how the performance varies with video length on Epic-Kitchens-100 videos in the Pre-training setting. Specifically, by 10th task, ~30 hours of video length is processed, by 20th task ~50 hours, and by 30th task ~80 hours. As seen below, the method’s performance remains consistent with increasing video length.
>
> Incremental Only Setting (described in 5.2): Average training (Train) and evaluation (Eval) accuracy at the end of task T on Epic Kitchens-100.
> | Setting            | Task  | 10   | 20   | 30  |
> |--------------------|---------|------|------|------|
> | Incremental    | Train.| 28.5 | 31.2 | 29.7 |
> |                        | Eval. | 27.5 | 24.6 | 32.3 |
>
> Since our method is a memory-based approach, the performance will degrade when the rehearsal buffer is unable to store data samples. Thus, due to reduced data samples from past tasks for rehearsal, forgetting may occur. To quantify the maximum video length beyond which our method’s performance degrades, we may also have to quantify an upper bound on the rehearsal buffer’s storage in Gb. As seen in Figure 4, SMILE [1] does not achieve stable performance under a limited memory budget, and in contrast, REMIND [2] requires 20 Gb for comparable performance in Kinetics-700. If one assumes 20 Gb as an upper bound, our method can process 5470 hours of video length.
>
> ---
>
> References:
> 1. SMILE: "Just a Glimpse: Rethinking Temporal Information for Video Continual Learning", CVPR 2023.
> 2. REMIND: "REMIND Your Neural Network to Prevent Catastrophic Forgetting". ECCV 2020.
>
> We are grateful for the chance to discuss our work's potential, and wish to thank you again for your valuable input.

---

### Meta-Review · Area_Chair_uMuH · 2024-12-21

**Metareview:**

The paper received mixed reviews. Two reviewers vote for borderline acceptance while the other two (especially ngdC) are firmly on the rejection side. The AC checked all the materials and concurs that the paper has done a reasonable exploration of continual learning with memories storing codes which are potentially helpful to address the catastrophic forgetting issue, and the authors have clarified concerns and improved the draft during the rebuttal and discussion process. However, even the borderline acceptance reviewer (h5dS) still remains concerned about paper writing, quoting "the paper still requires substantial revisions to improve its clarity before publication". Weighing all the factors, the AC decides the paper is not ready for publication and would require major revisions for the next cycle.

**Additional Comments On Reviewer Discussion:**

Please see the reasoning in the meta review.

Regarding writing clarity, the authors have made attempts to improve locally (e.g., related work, captions) as requested by the reviewers. However, multiple reviewers (h5dS and ngdC) believe the paper needs major, global revisions to be ready for publication.

---

### Decision · Program_Chairs · 2025-01-22

Reject